# Per-Architecture Training-Free Metric Optimization for Neural Architecture Search

**Mingzhuo Lin[1], Jianping Luo[1*]**
[1] Guangdong Key Laboratory of Intelligent Information Processing,
College of Electronic and Information Engineering, Shenzhen University, China

## Abstract

Neural Architecture Search (NAS) aims to identify high-performance networks within a defined search space. Training-free metrics have been proposed to estimate network performance without actual training, reducing NAS deployment costs. However, individual training-free metrics often capture only partial architectural features, and their estimation capabilities are different in various tasks. Combining multiple training-free metrics has been explored to enhance scalability across tasks. Yet, these methods typically optimize global metric combinations over the entire search space, overlooking the varying sensitivities of different architectures to specific metrics, which may limit the final architectures' performance. To address these challenges, we propose the Per-Architecture Training-Free Metric Optimization NAS (PO-NAS) algorithm. This algorithm: (a) Integrates multiple training-free metrics as auxiliary scores, dynamically optimizing their combinations using limited real-time training data, without relying on benchmarks; (b) Individually optimizes metric combinations for each architecture; (c) Integrates an evolutionary algorithm that leverages efficient predictions from the surrogate model, enhancing search efficiency in large search spaces. Notably, PO-NAS combines the efficiency of training-free search with the robust performance of training-based evaluations. Extensive experiments demonstrate the effectiveness of our approach. Our code has been made publicly available at `https://github.com/LMZ-Zhuo/PO-NAS`.

## 1 Introduction

Neural Architecture Search (NAS) has emerged as a method to automate architecture engineering. Currently, NAS methods have surpassed many manually designed architectures in various tasks, such as image classification (1), object detection (2), and semantic segmentation (3). While many training-based NAS algorithms have achieved state-of-the-art (SOTA) performance across various tasks, their search costs are often prohibitively high in resource-constrained scenarios, primarily due to the necessity of training deep neural networks during the search process. To address this, numerous training-free metrics have been developed to evaluate candidate architectures' performance (4; 5; 6; 7; 8; 9; 10), eliminating the need for actual training. For instance, SWAP-NAS (10) achieves SOTA performance on the ImageNet dataset within the DARTS (11) search space in just 9 minutes of search time. These metrics are typically computed through a single forward and backward pass using a small batch of data, rendering their computational overhead negligible compared to traditional NAS methods. The efficacy of training-free metrics is typically assessed by evaluating the correlation between the rankings they produce and the actual performance rankings of architectures (12). However, a significant limitation of these metrics lies in their transferability. Studies have demonstrated considerable variability in the consistency between scores derived from training-free metrics and actual performance rankings across different tasks (5; 13; 14; 15).

To address the limited transferability of individual training-free metrics in diverse tasks, researchers have explored combining multiple such metrics for architecture evaluation. Depending on the methods

of combination, several studies have emerged. Such as EZNAS (16), Auto-Prox (17), Auto-GAS (18) and Auto-DAS (18), combine various symbolic regression operators to optimize training-free metrics. These methods optimize the metrics by aligning the combined metrics scores with the performance rankings across multiple task benchmarks. However, these approaches typically rely on a substantial amount of training data. Moreover, the optimized training-free metrics are often tailored specifically for the tasks in the training benchmarks, thereby limiting their applicability to a broader range of search tasks. Some approaches aim to integrate training-free metrics with training-based search methods. OMNI (19), ProxyBO (20), HNAS (14), and RoBoT (15) utilize training-free metrics as auxiliary information rather than sole evaluation criteria. During the training process, these methods evaluate the significance of training-free metrics based on actual performance and refine these metrics through surrogate models. However, these methods typically optimize the metric weights over the entire search space, neglecting the varying sensitivities of different architectures to each metric. As a result, the correlation of training-free metrics varies significantly among architectures with different features, even within the same search space. Our experiments on NAS-Bench-201 (21) across various architecture sets confirm this point, with detailed results in Appendix C.1.

To address the aforementioned challenges, we propose PO-NAS. PO-NAS assigns different weights to training-free metrics for each architecture, uses the weighted sum of metrics for scoring, and employs this score as a surrogate model to select the optimal architecture. It also dynamically adjusts the weight allocation based on performance feedback, which enhances PO-NAS's transferability across diverse unknown tasks. Additionally, PO-NAS employs an evolutionary algorithm that efficiently utilizes the surrogate model's evaluation capabilities to enhance the exploration of the search space. Our main contributions are as follows:

- We propose a surrogate modeling method, which assigns training-free metric weights to each architecture, uses the weighted sum of these metrics as a surrogate model, and dynamically adjusts the weights according to performance feedback.
- We propose an evolutionary algorithm that employs the surrogate model to efficiently explore the search space. With particular parental selection mechanism along with crossover and mutation methods, it enhances the balance between exploration and exploitation.
- We propose PO-NAS, a NAS method that effectively combines the efficiency of training-free NAS with the robust performance of training-based NAS. Extensive experiments across various datasets demonstrate PO-NAS's superior performance and scalability.

## 2 Related Work

**Training-free NAS**   Recent studies have proposed numerous training-free metrics for evaluating the performance of different network architectures. Model-dependent training-free metrics are designed for specific types of network architectures. For example, Zen-NAS (22) uses the Gaussian complexity of linear classifiers within each linear region to measure the distribution of linear regions. On the other hand, model-independent metrics exhibit higher transferability across different neural network structures (23). For example, NASI (22) leverages the ability of Neural Tangent Kernel (NTK) (24) to characterize architecture performance at initialization to develop new training-free metrics. However, many metrics still lack scalability across various architectures and tasks. Recent metrics such as Zico (8), SWAP (10), and AZ-NAS (9) have demonstrated excellent performance across various search tasks, but they still struggle to consistently exhibit robust evaluation performance across multiple tasks. Recent research indicates that training-free metrics are generally complementary to one another, an appropriate combination of these metrics can enhance rank correlation across multiple tasks (13).

**Hybrid NAS**   Some studies have attempted to integrate training-free metrics with training-based neural architecture search methods to better balance efficiency and performance. These approaches use the ground-truth performance on the target task as a supervisory signal. For example, OMNI (19) accelerates architecture search by combining training-free metrics with Bayesian Optimization, while ProxyBO (20) incorporates these metrics as input to the surrogate model to improve search efficiency. HNAS (14) introduces techniques such as gradient normalization and dynamic weight adjustment to improve the stability and reliability of gradient-based metrics. However, this approach remains limited by the intrinsic constraints of single metrics. RoBoT (15) employs a weighted linear combination to ensemble multiple training-free metrics and uses Bayesian Optimization to dynamically adjust their weights based on real-time training feedback. However, these methods

typically optimize the metric weights over the entire search space, neglecting the varying sensitivities of different architectures to each metric. They often settle for suboptimal solutions and struggle to identify the globally optimal architecture. In contrast to these methodologies, PO-NAS analyzes the sensitivity of individual architectures to various training-free metrics, assigning distinct metric weights for each architecture accordingly, which enhances PO-NAS's evaluation capability.

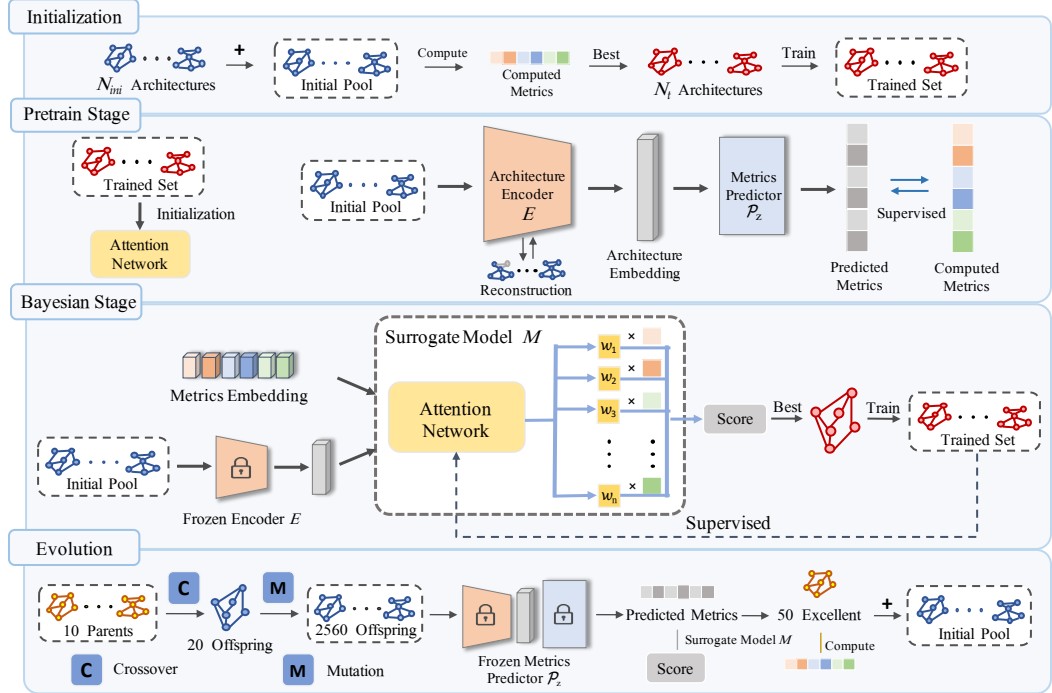

Figure 1: Overview of PO-NAS. First, we randomly initialize a large number of architectures as the initial pool and select several architectures for training to form the trained set. Pre-training stage, we initialize the attention network and optimize the architecture encoder as well as the metrics predictor with training-free metrics to generate embeddings that distinguish between different architectures. Bayesian stage, we use the surrogate model to search for the optimal architecture, and optimize the network which assigns weights to metrics for each architecture based on performance feedback. We combine the surrogate model with the evolutionary algorithm to extensively explore the search space.

## 3 Our Methodology

### 3.1 Algorithm Principle

An appropriate combination of training-free metrics can enhance rank correlation across multiple tasks (13). In this section, we explain how PO-NAS leverages training-free metrics to select the best architecture while using the true performance feedback to optimize the metric weights. Let the search space be defined as a set containing $N$ candidate architectures, i.e., $\mathbb{A} = \{\mathcal{A}_i\}_{i=1}^{N}$. The available information includes: (a) the architecture graph $\mathcal{G}(\mathcal{A})$; (b) the training-free metrics $\mathcal{Z}(\mathcal{A})$ and (c) the true performance $f(\mathcal{A})$. When true performance data is limited, it is essential to fully exploit the available information. $\mathcal{G}(\mathcal{A})$ can be obtained as a prior; $\mathcal{Z}(\mathcal{A})$ can be computed at a low cost; whereas $f(\mathcal{A})$ is the most valuable but also the most computationally expensive, making it difficult to serve as large-scale supervision. Thus, we employ $\mathcal{Z}(\mathcal{A})$ and $\mathcal{G}(\mathcal{A})$ to train the architecture encoder, enabling it to generate embeddings that distinguish architectures with different characteristics. This aspect will be discussed in detail in Section 3.3. We assign different metric weights for each architecture based on performance feedback $f(\mathcal{A})$. Given the constraints of limited search budgets $T$, we adopt a weighted linear combination as the fusion strategy for the metrics. This formulation is not only more robust but also more efficient to optimize under limited supervision. Let $\mathcal{Z}_1(\mathcal{A}), \mathcal{Z}_2(\mathcal{A}), \ldots, \mathcal{Z}_k(\mathcal{A})$

be $K$ complementary metrics. We define the scoring function for a single architecture $\mathcal{A}$ as:

$$\mathcal{S}(\mathcal{A}; w_{\mathcal{A}}) = \sum_{i=1}^{k} w_{\mathcal{A},i} \cdot \tilde{\mathcal{Z}}_i(\mathcal{A}) \tag{1}$$

where $w_{\mathcal{A}} \in \Delta^k$ is the weight vector corresponding to the architecture (satisfying $\sum |w_{\mathcal{A},i}| = 1$), and $\tilde{\mathcal{Z}}_i(\mathcal{A})$ denotes the normalized value of the $i$-th metric. To maximize the rank correlation between the true performance $f(\mathcal{A})$ and the combined score $\mathcal{S}(\mathcal{A}; w_{\mathcal{A}})$ over a validation set, we formulate the following weight optimization objective:

$$w_{\mathcal{A}}^{(t+1)} = \arg\max_{w_{\mathcal{A}}} \tau \left( \left\{ \mathcal{S}(\mathcal{A}; w_{\mathcal{A}}^{(t)}) \right\}_{\mathcal{A} \in \mathbb{A}_t}, \{f(\mathcal{A})\}_{\mathcal{A} \in \mathbb{A}_t} \right) \tag{2}$$

where $\tau(\cdot)$ denotes the Kendall rank correlation coefficient, and $\mathbb{A}_t$ is the set of candidate architectures selected in iteration $t$. With the optimized weights $w_{\mathcal{A}}^{(t)}$, we can evaluate both the entire set of candidates $\mathbb{A}_t$ and the architectures discovered through the evolutionary algorithm.

### 3.2 Overall Framework

---

**Algorithm 1** Pseudo code for PO-NAS

---

**Require:** True training performance function $f$, true training-free metrics function $\mathcal{Z}$, the number of the candidate architecture $N_{ini}$, the number of the initial trained architecture $N_t$, the pre-training epochs $T_p$, the BO search budget $T_s$ and the evolution iteration $T_e$
 1: // Initialization (Appendix B.3)
 2: Candidate architecture set $\mathbb{A}_0 \leftarrow$ Generate randomly $N_{ini}$ architectures
 3: Candidate metrics set $\mathbb{Z}_0 \leftarrow$ Compute the real training-free metrics $\mathcal{Z}(\mathbb{A}_0)$
 4: Trained set $\mathcal{Q}_0 \leftarrow$ Select $N_t$ architectures from $\mathbb{A}_0$ and obtain $(\mathbb{A}_{ini}, f(\mathbb{A}_{ini}))$ by training $\mathbb{A}_{ini}$
 5: // Pre-training stage (Section 3.3)
 6: **for** step $t = 1, \ldots, T_p$ **do**
 7:     Train the architecture encoder and the metrics predictor using $\mathbb{Z}_0$ and $\mathbb{A}_0$
 8:     Optimize the architecture encoder with the node feature masking reconstruction task.
 9: **end for**
10: Obtain architecture encoder $E$ and metrics predictor $\mathcal{P}_z$
11: // BO search stage (Section 3.4)
12: **for** step $t = 1, \ldots, T_s$ **do**
13:     Update the surrogate model $M$ using $\mathcal{Q}_{t-1}$
14:     **if** $t > T_e$ **then**
15:         // Evolution stage (Section 3.5)
16:         Choose the excellent architecture set $\mathbb{A}_{ex}$ by $M(E(\mathbb{A}_{t-1}), \mathbb{Z}_{t-1})$ from $\mathbb{A}_{t-1}$
17:         Select excellent architecture combinations $\mathbb{C}_{ex}$ by $\mathcal{S}_{pair}$(Appendix A.4) from $\mathbb{A}_{ex}$
18:         Obtain the offspring set $\mathbb{A}_{ch}$ by applying crossover and mutation operations to $\mathbb{C}_{ex}$
19:         Choose excellent offspring architectures $\mathbb{A}_{ch_{ex}}$ by $M(E(\mathbb{A}_{ch}), \mathcal{P}_z(\mathbb{A}_{ch}))$ from $\mathbb{A}_{ch}$
20:         Compute the real training-free metrics $\mathcal{Z}(\mathbb{A}_{ch_{ex}})$
21:         Obtain $\mathbb{A}_t = \mathbb{A}_{t-1} \cup \{\mathbb{A}_{ch_{ex}}\}$ and $\mathbb{Z}_t = \mathbb{Z}_{t-1} \cup \{\mathcal{Z}(\mathbb{A}_{ch_{ex}})\}$
22:     **end if**
23:     Obtain the best architecture $\mathcal{A}_{best}$ by $M(E(\mathbb{A}_t), \mathbb{Z}_t)$ from $\mathbb{A}_t$
24:     **while** $\mathcal{A}_{best} \in \mathcal{Q}_{t-1}$ **do**
25:         Select next best architecture $\mathcal{A}_{next}$ as $\mathcal{A}_{best}$
26:     **end while**
27:     Obtain the true performance $f(\mathcal{A}_{best})$ by training $\mathcal{A}_{best}$
28:     Obtain $\mathcal{Q}_t = \mathcal{Q}_{t-1} \cup \{(\mathcal{A}_{best}, f(\mathcal{A}_{best}))\}$
29: **end for**
30: **return** the best architecture $\mathcal{A}_{best}^* = \arg\max_{\mathcal{A}} f(\mathcal{A})_{(\mathcal{A}, f(\mathcal{A})) \in \mathcal{Q}_{T_s}}$

---

Bayesian Optimization (25) can effectively refine the surrogate model within a limited training budget, making it highly suitable for our algorithm. Thus, we employ Bayesian Optimization (BO) to optimize our surrogate model. The overall framework of PO-NAS is presented in Figure 1.

Algorithm 1 describes our proposed method. (a) Initialization (lines 1-4 of Algorithm 1): We randomly initialize a large number of architectures as the candidate set and select several architectures for training based on the average of all training-free metrics to form the initial trained set (Implementation details can be found in Appendix B.3); (b) Pre-training stage (lines 5-10 of Algorithm 1): We optimize the architecture encoder and the metric predictor using the initial architecture set and their training-free metrics (Section 3.3); (c) BO search stage (lines 11-29 of Algorithm 1): At each iteration, we first update our surrogate model with the actual performance data of the trained architectures. Once the surrogate model has been trained to a certain loss threshold, we use it to predict the performance score of each architecture, rank the architecture pool based on these scores, select the highest-scoring architecture for actual training, and then update the pool of trained architectures (Section 3.4); (d) Evolution stage (lines 15-21 of Algorithm 1): We select advantageous offspring based on predicted scores, calculate the operation costs between them, and choose the best combinations for crossover and mutation to obtain the offspring set (lines 16-18 of Algorithm 1). Additionally, we use the surrogate model to forecast the scores of the offspring and select the top-ranked ones to calculate their training-free metrics and update the initial architecture pool (lines 19-21 of Algorithm 1) (Section 3.5). After several cycles of iteration, we identify the optimal architecture.

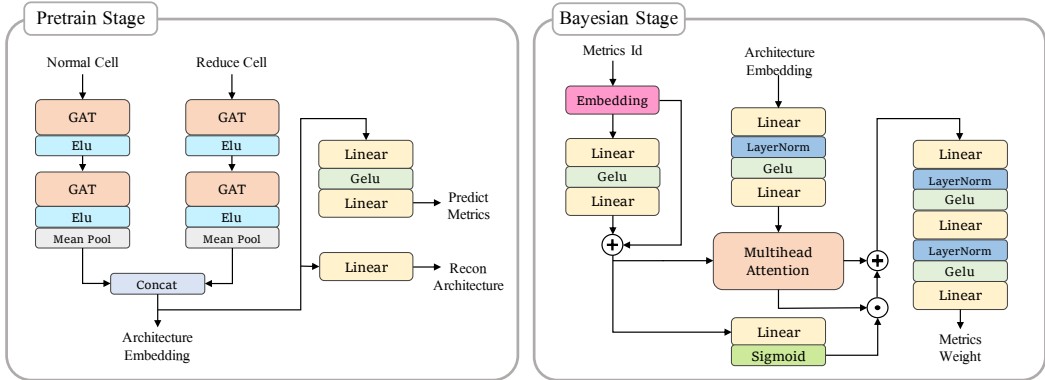

Figure 2: Model design of PO-NAS. Pre-training stage includes an architecture encoder, as well as metric predictor and architecture reconstruction heads for supervised training. Bayesian stage includes an attention network, which is utilized to generate metric weights by integrating architectural embeddings with metric embeddings.

## 3.3 Architecture Encoder

In this section, we will provide a detailed description of the Architecture Encoder component. The purpose of this component is to learn the structural characteristics of architectures through training-free metrics, thereby generating embeddings that can distinguish between different types of architectures. We employ a multi-head graph attention mechanism (GAT) (26) to perform hierarchical feature learning on the neural architecture. The model design of the architecture encoder is shown in Figure 2. We use a two-layer GAT to encode graph features. We employ the Exponential Linear Unit (ELU) post the GAT, driven by the nature of node connectivity in the topology. GAT processes connections and operations between nodes, some of which may negatively correlate with predictions. ELU's smooth negative region $(a(e^x - 1))$ retains these critical negative features. The final graph-level embedding $h_{\mathcal{G}}$ is obtained through global average pooling. To effectively learn the expansion structure of architectures, inspired by GraphMAE (27), we design a node feature masking reconstruction task to enhance the robustness of architectural features. Specifically, we randomly mask node features $x_v$ with a probability $p_{\text{mask}}$, replace them with learnable mask tokens $m$, and reconstruct the original features through a linear decoder. The reconstruction loss is defined as follows:

$$\mathcal{L}_{\text{recon}} = \frac{1}{|\mathcal{V}_m|} \sum_{v \in \mathcal{V}_m} \|\hat{x}_v - x_v\|_2^2 \tag{3}$$

Where $\mathcal{V}_m$ represents the set of masked nodes, and $\hat{x}_v$ is the reconstruction result. We define the metric prediction loss as follows:

$$\mathcal{L}_{\text{metric}} = \frac{1}{k} \sum_{i=1}^{k} \left\| \mathcal{P}_z^i(h_\mathcal{G}) - \mathcal{Z}_i(\mathcal{G}) \right\|_2^2 \tag{4}$$

Here, $\mathcal{P}_z^i(h_\mathcal{G})$ represents the $i$-th metric prediction value, and $\mathcal{Z}_i(\mathcal{G})$ is the $i$-th training-free metric value. During the pre-training phase, the encoder is optimized through the self-supervised reconstruction loss and the training-free metric prediction loss. After the pre-training phase, we freeze the architecture encoder and metric predictor, providing the graph-level embedding $h_\mathcal{G}$ and the metric prediction values $\mathcal{P}_z^i(h_\mathcal{G})$ for subsequent search stages.

### 3.4 Surrogate Model

**Model Design**    We design a surrogate model based on a multi-head cross-attention network (28), aimed at assigning training-free metric weights for each architecture and generating architecture scores. The design of the attention network is shown in Figure 2. We utilize the multi-head cross-attention mechanism to establish associations between architectural embeddings and metric embeddings. Based on this, the network generates metric weights for each architecture, enabling the surrogate model to generate scores for predicting the performance of each architecture. We introduce a dynamic gating module to regulate the integration ratio between attention outputs and original metric embeddings. To improve cross-modal interaction efficiency, each metric embedding and architectural embedding is equipped with a Linear-GELU-Linear trio, mapping the original embeddings into a higher-dimensional space. We employ GELU to enhance the representation of high-level topological features. The linearly transformed features require refined nonlinear processing. GELU's probabilistic gating ($x \cdot \Phi(x)$) captures complex node interactions, while its smoothness aids hierarchical feature propagation. Each metric is transformed into a dense vector through an embedding layer, with residual connections incorporated to retain original features and enhance representational capacity. A three-layer MLP network is employed to generate metric weights.

**Normalize and Loss Function**    To fairly evaluate the sensitivity of different architectures to various metrics, we normalize the training-free metrics and their weights. The specific implementation details can be found in Appendix B.3. After obtaining the normalized metrics $\hat{\mathcal{Z}}_i$ and the weights $\hat{w}_i$, we decompose the normalized weights $\hat{w}$ into positive activation weights $\hat{w}^+ = \text{ReLU}(\hat{w})$ and negative activation $\hat{w}^- = \text{ReLU}(-\hat{w})$, and construct the scoring function as follows:

$$\hat{\mathcal{S}} = \sum_{i=1}^{k} \left( \hat{w}_i^+ \cdot \hat{\mathcal{Z}}_i + \hat{w}_i^- \cdot (1 - \hat{\mathcal{Z}}_i) \right) \tag{5}$$

Most studies evaluate the performance of training-free metrics by calculating the correlation coefficient between the ranking of training-free metrics and the actual performance ranking of architectures. However, in this algorithm, due to the limited availability of real performance data, calculating the correlation coefficient is not feasible. Therefore, we use the alignment loss between the distribution of differences in architecture scores and actual performance as the main supervisory signal for the surrogate model. Due to the model score being ultimately derived from a weighted sum of metrics, it is challenging to fine-tune the weights to ensure accurate ranking among architectures with closely matched performance. Hence, we introduce a difference threshold $\mathcal{T}_{\text{th}}$ to reduce the impact of architectures with closely similar performance on the optimization difficulty. We calculate the loss only for correctly ranked architecture combinations. For the trained set $\mathcal{Q}_t$ of iteration $t$, where the true performance of the architecture has been obtained, the alignment loss is defined as follows:

$$\delta_{\text{pred}} = \hat{\mathcal{S}}_{\mathcal{A}_i} - \hat{\mathcal{S}}_{\mathcal{A}_j}, \quad \delta_{\text{true}} = f(\mathcal{A}_i) - f(\mathcal{A}_j) \quad (\forall i, j \in C(\mathcal{Q}_t, 2))$$

$$\mathcal{M} = \begin{cases} 1, & \text{if } |\delta_{\text{true}}| < T_{\text{th}} \\ 1, & \text{if } \delta_{\text{pred}} \cdot \delta_{\text{true}} > 0 \\ 0, & \text{otherwise} \end{cases}$$

$$\mathcal{L}_{\text{align}} = \mathbb{E} \left[ \left\| \frac{|\delta_{\text{pred}}| \odot \mathcal{M} - \mathbb{E}\left[|\delta_{\text{pred}}|\right]}{\sigma(|\delta_{\text{pred}}|)} - \frac{|\delta_{\text{true}}| \odot \mathcal{M} - \mathbb{E}\left[|\delta_{\text{true}}|\right]}{\sigma(|\delta_{\text{true}}|)} \right\| \right] \tag{6}$$

$\mathbb{E}(\cdot)$ denotes the expectation, $\sigma(\cdot)$ denotes the standard deviation and $\odot$ denotes the element-wise multiplication. Additionally, we incorporate the correlation between scores and performance $\mathcal{L}_{\text{corr}}$ as an auxiliary loss and add a direction alignment penalty $\mathcal{L}_{\text{dir}}$ to ensure overall ranking accuracy:

$$\mathcal{L}_{\text{corr}} = 1 - \rho \left( \left\{ \hat{\mathcal{S}}_{\mathcal{A}} \right\}_{\mathcal{A} \in \mathcal{Q}_t}, \{f(\mathcal{A})\}_{\mathcal{A} \in \mathcal{Q}_t} \right) \tag{7}$$

where $\rho(\cdot)$ denotes the Pearson correlation coefficient.

$$\mathcal{L}_{\text{dir}} = \mathbb{E} \left[ \text{ReLU}(-\delta_{\text{pred}} \cdot \delta_{\text{true}}) \right] \tag{8}$$

At the beginning of each iteration, the surrogate model is updated through the alignment loss, the correlation loss, and the direction alignment loss.

## 3.5 Evolutionary Algorithm

We develop a novel evolutionary algorithm that employs the surrogate model to efficiently explore the search space. Detailed methods are provided in Appendix A. This algorithm employs matrix encoding to map topological structures into operable matrix forms. To effectively measure the intrinsic differences between architectures, we compute the shortest operation path and operation cost between two parent architectures and propose the shortest operation path crossover.

We enumerate the neighborhood space of elite architectures using a neighborhood mutation strategy and adopt an adaptive selection mechanism to balance exploration and exploitation. As shown in Figure 3, in the early stages of search, it prioritizes offspring combinations with higher computational costs to enhance global search capabilities by increasing architectural diversity. In the later stages, it shifts to fitness-based selection, focusing on high-performing individuals to deepen local optimization. The algorithm leverages the efficient evaluation speed of the surrogate model described in Section 3.4 to efficiently explore the search space. Through the evolutionary algorithm, as illustrated in Figure 1, we are able to focus on a large number of samples in the search space with very limited computational cost, thereby identifying optimal solutions among the search space.

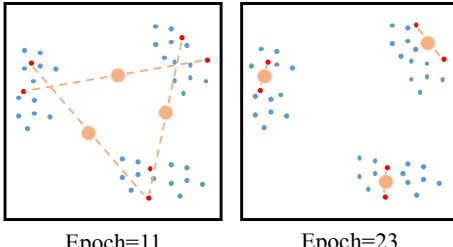

Epoch=11          Epoch=23

Figure 3: These figures represent two-dimensional (2-D) embeddings of high-dimensional search space. Red dots: superior architectures; Blue dots: candidate architectures; Orange dots: exploration areas.

# 4 Experiments

## 4.1 PO-NAS on NAS Benchmark

To validate the robustness and exceptional predictive performance of PO-NAS across various tasks, we conduct comprehensive experiments on multiple popular NAS benchmarks. These benchmarks include 20 distinct training tasks: NAS-Bench-201 (21), TransNAS-Bench-101 (34), and DARTS (11). PO-NAS utilizes 6 metrics outlined by (5): grad_norm, snip, grasp, fisher, synflow, and jacob_cov. Implementation details and more empirical results can be found in Appendix B and Appendix C.

Table 1, Table 2, Table 3 and Table 4 present the performance results of PO-NAS on NAS-Bench-201, DARTS and TransNAS-Bench-101. In the large-scale DARTS search space, PO-NAS demonstrates remarkable performance. This is attributed to its integration of the advantages of both training-free and training-based NAS methods. It optimizes training-free metrics based on training performance without relying on training benchmarks and leverages the efficient computational speed of training-free metrics. By utilizing training-free metrics and the layer-wise evaluation of the surrogate model, PO-NAS only requires training a small number of architectures (25), yet it can focus on a large number of architectures (approximately 50000) during the search phase. With a balanced exploration-exploitation search strategy, PO-NAS can rapidly and effectively explore the large-scale search space relying solely on limited training. Moreover, the results on NAS-Bench-201 and TransNAS-Bench-101 indicate that PO-NAS significantly enhances the evaluation capability of the original metrics. By optimizing the corresponding metric combinations for different types of architectures, PO-NAS gains

Table 1: Comparison of various NAS algorithms in NAS-Bench-201. Results are reported with the mean ± standard deviation of 10 runs. "Training-free (Avg./Best)" represent the average value of metrics and the best individual metrics. The best results are **bold**, and the second best are underlined.

| Algorithm | Test Accuracy (%) | | | Cost (GPU Sec.) | Method |
|---|---|---|---|---|---|
| | C10 | C100 | IN-16 | | |
| ResNet (29) | 93.97 | 70.86 | 43.63 | - | manual |
| REA[†] | 93.92±0.30 | 71.84±0.99 | 45.15±0.89 | 12000 | evolution |
| RS (w/o sharing)[†] | 93.70±0.36 | 71.04±1.07 | 44.57±1.25 | 12000 | random |
| REINFORCE[†] | 93.85±0.37 | 71.71±1.09 | 45.24±1.18 | 12000 | RL |
| BOHB[†] | 93.61±0.52 | 70.85±1.28 | 44.42±1.49 | 12000 | BO+bandit |
| DARTS (2nd) (11) | 54.30±0.00 | 15.61±0.00 | 16.32±0.00 | 43277 | gradient |
| DrNAS (30) | 93.98±0.58 | 72.31±1.70 | 44.02±3.24 | 14887 | gradient |
| Sharply-NAS (31) | 94.05±0.19 | 73.15±0.26 | 46.25±0.25 | 14762 | gradient |
| $\beta$-DARTS (32) | 94.00±0.22 | 72.91±0.43 | 46.20±0.38 | 3280 | gradient |
| TE-NAS (4) | 93.90±0.47 | 71.24±0.56 | 42.38±0.46 | 1558 | training-free |
| NASI (33) | 93.55±0.10 | 71.20±0.14 | 44.84±1.41 | 120 | training-free |
| GradSign (7) | 93.31±0.47 | 70.33±1.28 | 42.42±2.81 | 1824 | training-free |
| ZiCo (8) | 93.50±0.18 | 70.62±0.26 | 42.04±0.82 | 372 | training-free |
| AZ-NAS (9) | 93.53±0.15 | 70.75±0.48 | 45.43±0.29 | 43 | training-free |
| HNAS (14) | 94.04±0.21 | 71.75±1.04 | 45.91±0.88 | 3010 | hybrid |
| RoBoT (15) | **94.36±0.00** | **73.51±0.00** | 46.34±0.00 | 3051 | hybrid |
| Training-free (Avg.) | 92.00 | 70.66 | 45.73 | - | training-free |
| Training-free (Best) | 93.76 | 71.11 | 42.60 | - | training-free |
| PO-NAS | 94.12±0.22 | **73.51±0.00** | **46.71±0.12** | 3162 | hybrid |
| Optimal | 94.37 | 73.51 | 47.31 | - | - |

[†]Reported by (21)

Table 2: Performance comparison among various NAS algorithms on ImageNet on DARTS search space. The search costs are evaluated on an Nvidia 1080Ti. The best results are in **bold**, and the second best are underlined.

| Algorithm | Test Error (%) | | Params (M) | +× (M) | Search Cost (GPU Days) |
|---|---|---|---|---|---|
| | Top-1 | Top-5 | | | |
| Inception-v1 (35) | 30.1 | 10.1 | 6.6 | 1448 | - |
| MobileNet (36) | 29.4 | 10.5 | 4.2 | 569 | - |
| AmoebaNet-A (1) | 25.5 | 8.0 | 5.1 | 555 | 3150 |
| PNAS (37) | 25.8 | 8.1 | 5.1 | 588 | 225 |
| MnasNet-92 (38) | 25.2 | 8.0 | 4.4 | 388 | - |
| DARTS (11) | 26.7 | 8.7 | 4.7 | 574 | 4.0 |
| ProxylessNAS (39) | 24.9 | 7.5 | 7.1 | 465 | 8.3 |
| SDARTS-ADV (40) | 25.2 | 7.8 | 5.4 | 594 | 1.3 |
| TE-NAS (4) | 24.5 | 7.5 | 5.4 | - | 0.17 |
| NASI-ADA (33) | 25.0 | 7.8 | 4.9 | 559 | 0.01 |
| QE-NAS (41) | 25.5 | - | 3.2 | - | 0.02 |
| SWAP-NAS (10) | 24.0 | 7.6 | 5.8 | - | 0.006 |
| HNAS (14) | 24.3 | 7.4 | 5.1 | 575 | 0.5 |
| RoBoT (15) | 24.1 | 7.3 | 5.0 | 556 | 0.6 |
| PO-NAS | **23.9** | **7.1** | 6.3 | 667 | 0.64 |

an advantage in exploring the best architecture. However, this optimization also correspondingly increases the difficulty of surrogate model optimization, thereby affecting the stability of its performance to some extent. Nevertheless, PO-NAS still achieves competitive results.

Table 3: Performance comparison among various NAS algorithms on CIFAR-10/100 on DARTS search space. The performance of the final architectures selected by PO-NAS is reported with the mean ± standard deviation of 5 runs. The search costs are evaluated on an Nvidia 1080Ti. The best results are in **bold**, and the second best are underlined.

| Algorithm | Test Error (%) | | Params (M) | | Search Cost (GPU Hours) | Search Method |
|---|---|---|---|---|---|---|
| | C10 | C100 | C10 | C100 | | |
| DenseNet-BC (42) | 3.46* | 17.18* | 25.6 | 25.6 | - | manual |
| NASNet-A (43) | 2.65 | - | 3.3 | - | 48000 | RL |
| AmoebaNet-A (1) | 3.34±0.06 | 18.93† | 3.2 | 3.1 | 75600 | evolution |
| PNAS (37) | 3.41±0.09 | 19.53* | 3.2 | 3.2 | 5400 | SMBO |
| ENAS (44) | 2.89 | 19.43* | 4.6 | 4.6 | 12 | RL |
| NAONet (45) | 3.53 | - | 3.1 | - | 9.6 | NAO |
| DARTS (2nd) (11) | 2.76±0.09 | 17.54† | 3.3 | 3.4 | 24 | gradient |
| GDAS (46) | 2.93 | 18.38 | 3.4 | 3.4 | 7.2 | gradient |
| NASP (47) | 2.83±0.09 | - | 3.3 | - | 2.4 | gradient |
| DARTS- (avg) (11) | 2.59±0.08 | 17.51±0.25 | 3.5 | 3.3 | 9.6 | gradient |
| SNAS (48) | 2.85±0.02 | 20.09 | 2.8 | 2.8 | 36 | gradient |
| SETN (49) | 2.69 | 17.25 | 3.4 | 3.4 | 43.2 | gradient |
| SDARTS-ADV (40) | 2.61±0.02 | - | 3.3 | - | 31.2 | gradient |
| R-DARTS (L2) (50) | 2.95±0.21 | 18.01±0.26 | - | - | 38.4 | gradient |
| TE-NAS (4) | 2.83±0.06 | 17.42±0.56 | 3.8 | 3.9 | 1.2 | training-free |
| NASI-ADA (33) | 2.90±0.13 | 16.84±0.40 | 3.7 | 3.8 | 0.24 | training-free |
| HNAS (14) | 2.62±0.04 | **16.29±0.14** | 3.4 | 3.8 | 2.6 | hybrid |
| RoBoT (15) | 2.60±0.03 | 16.52±0.10 | 3.3 | 3.8 | 3.5 | hybrid |
| PO-NAS | **2.52±0.03** | 16.35±0.12 | 3.8 | 4.2 | 3.9 | hybrid |

## 4.2 Ablation Studies

We conduct a series of ablation studies on PO-NAS to investigate the impact of different components and parameter settings. These factors include ensemble method, pre-training loss, difference threshold $\mathcal{T}_{\text{th}}$ and loss threshold, the number of combined training-free metrics and evolutionary algorithm. Implementation details and more detailed experimental results can be found in Appendix D.

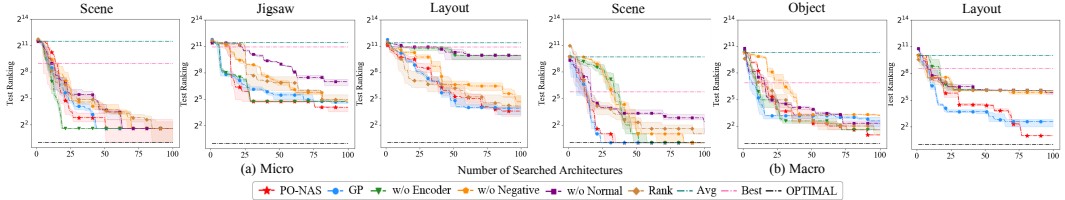

Figure 4: Ablation study of the surrogate model in TransNAS-Bench-101 regarding the number of searched architectures. All methods are reported with the mean and standard error of 10 independent searches. Implementation details and more empirical results are provided in Appendix D.1.

**Ablation Studies on Surrogate Model** To investigate the impact of the surrogate model and its various components on the performance of PO-NAS, we conduct a series of ablation studies on PO-NAS in TransNAS-Bench-101. These experiments focus on three main aspects: (a) Ablation studies of global optimization versus per-architecture optimization (GP, w/o encoder); (b) The influence

Table 4: Performance comparison of NAS algorithms in TransNAS-Bench-101. All methods search for 100 architectures. The results of RoBoT, HNAS and PO-NAS are reported with the mean ± standard deviation of 10 runs, while 50 independent searches for REA, RS, and REINFORCE. "Training-free (Avg./Best)" represent the average value of metrics and the best individual metrics.

| Space | Algorithm | Accuracy (%) | | | L2 Loss ($\times 10^{-2}$) | mIoU (%) | SSIM ($\times 10^{-2}$) | |
| --- | --- | --- | --- | --- | --- | --- | --- | --- |
| | | Scene | Object | Jigsaw | Layout | Segment. | Normal | Autoenco. |
| Micro | REA | 54.63±0.18 | 44.88±0.31 | 94.73±0.14 | -62.02±0.59 | 94.56±0.02 | 56.76±0.33 | **56.00±0.71** |
| | RS (w/o sharing) | 54.53±0.19 | 44.76±0.45 | 94.61±0.29 | -62.17±0.98 | 94.52±0.03 | 56.45±0.25 | 55.27±0.90 |
| | REINFORCE | 54.49±0.19 | 44.64±0.38 | 94.69±0.16 | -61.61±0.91 | 94.53±0.04 | 57.06±0.35 | 55.47±0.72 |
| | HNAS | 54.29±0.09 | 44.08±0.00 | 94.56±0.21 | -64.83±1.69 | 94.57±0.00 | 56.88±0.00 | 48.66±0.00 |
| | RoBoT | 54.87±0.00 | **45.59±0.00** | 94.82±0.06 | **-61.16±0.86** | 94.58±0.00 | **57.44±0.34** | 55.42±1.05 |
| | Training-free (Avg.) | 49.10 | 40.85 | 83.49 | -73.79 | 94.43 | 53.42 | 35.56 |
| | Training-free (Best) | 53.72 | 41.78 | 91.08 | -70.25 | 94.53 | 55.22 | 41.77 |
| | PO-NAS | **54.90±0.03** | **45.59±0.00** | **94.91±0.09** | -61.55±0.75 | **94.60±0.01** | 57.08±0.14 | 55.62±0.41 |
| | Optimal | 54.94 | 45.59 | 95.37 | -60.10 | 94.61 | 58.73 | 57.72 |
| Macro | REA | 56.65±0.31 | 46.87±0.33 | 96.75±0.08 | -60.38±1.12 | 94.80±0.03 | 60.57±0.56 | 71.39±2.76 |
| | RS (w/o sharing) | 56.69±0.25 | 46.60±0.36 | 96.72±0.24 | -60.43±1.16 | 94.76±0.03 | 60.61±0.51 | 71.07±2.75 |
| | REINFORCE | 56.43±0.29 | 46.66±0.30 | 96.80±0.14 | -60.36±1.11 | 94.78±0.04 | 60.34±0.52 | 69.21±2.55 |
| | HNAS | 55.03±0.00 | 45.00±0.00 | 96.28±0.18 | -61.40±0.11 | 94.79±0.00 | 59.27±0.00 | 57.59±0.00 |
| | RoBoT | 57.35±0.13 | 46.94±0.09 | 96.92±0.02 | -58.88±0.70 | 94.85±0.02 | 61.66±0.00 | 73.53±0.06 |
| | Training-free (Avg.) | 54.50 | 43.93 | 95.20 | -65.06 | 94.54 | 60.93 | 64.78 |
| | Training-free (Best) | 56.27 | 46.05 | 96.51 | -63.41 | 94.70 | 60.93 | 65.11 |
| | PO-NAS | **57.41±0.00** | **47.05±0.01** | **96.97±0.04** | **-58.44±0.22** | **94.86±0.00** | **64.35±0.00** | **74.19±0.23** |
| | Optimal | 57.41 | 47.42 | 97.02 | -58.22 | 94.86 | 64.35 | 74.88 |

of different normalization methods (w/o Negative, w/o Normal, Rank); (c) Ablation studies of the surrogate model itself: Eliminate the surrogate model and directly use the average value of all metrics (Avg) and the best individual metrics (Best) for prediction. "OPTIMAL" denotes the performance of the optimal architecture on the entire dataset. Partial experimental results are presented in Figure 4, which fully demonstrate the effectiveness of our surrogate model and its various components.

**Ablation Studies on Evolutionary Algorithm** We analyze the impact on PO-NAS performance from: (a) The evolutionary algorithm itself (w/o Evolution, REA); (b) The evolutionary algorithm components (w/o $N$, w/o Mutation, w/o Crossover); Figure 5 shows partial experimental results, indicating our evolutionary algorithm's robust and excellent search capability across various initial architecture counts. Implementation details and more empirical results are provided in Appendix D.2.

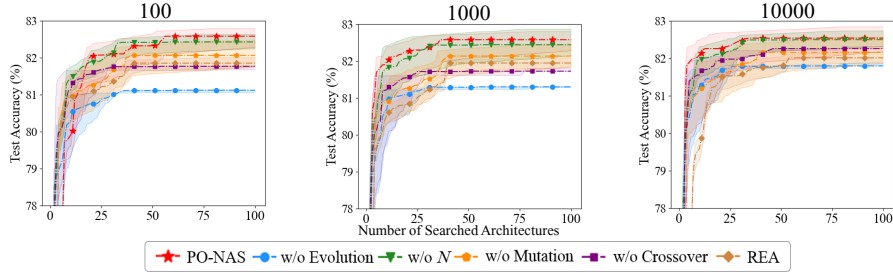

Figure 5: Ablation study of the evolutionary algorithm on DARTS regarding the number of searched architectures. All methods are reported with the mean and standard error of 10 independent searches.

## 5 Conclusion and Future Work

We propose PO-NAS which: (a) Employs a surrogate model to assign training-free metric weights for each architecture and generate architecture scores, effectively enhancing the capability to select the best architecture using training-free metrics; (b) Employs a novel evolutionary algorithm that integrates the surrogate model for rapid architecture evaluation to efficiently explore large-scale search spaces. Extensive experiments have demonstrated PO-NAS's superior performance and scalability. To enhance model stability, we plan to cluster architectures and optimize metrics for each cluster rather than for individual architectures in the future.

# 6 Acknowledgement

This work was supported by the National Natural Science Foundation of China under Grant 62176161, and the Scientific Research and Development Foundations of Shenzhen under Grant JCYJ20220818100005011 and 20200813144831001.

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

# A More Details about Evolutionary Algorithm

In this section of the appendix, we provide a detailed explanation of the evolutionary search algorithm employed in PO-NAS. PO-NAS utilizes a cell-based search space, similar to that found in related work such as DARTS (11). Regarding the search strategy, PO-NAS selects parent architectures based on operation costs and the scores provided by the surrogate model. The search process involves a series of candidate offspring rather than a single network at each step. This population-based approach allows for broader coverage of the search space, thereby increasing the likelihood of discovering high-performance architectures. Although evolutionary search algorithms are typically resource-intensive due to the need for multiple evaluations, PO-NAS mitigates this drawback by leveraging the low assessment costs of the surrogate model. Thus, our aim is to offer an evolutionary algorithm that is both efficient and effective, enabling extensive exploration of the search space without incurring excessive computational costs.

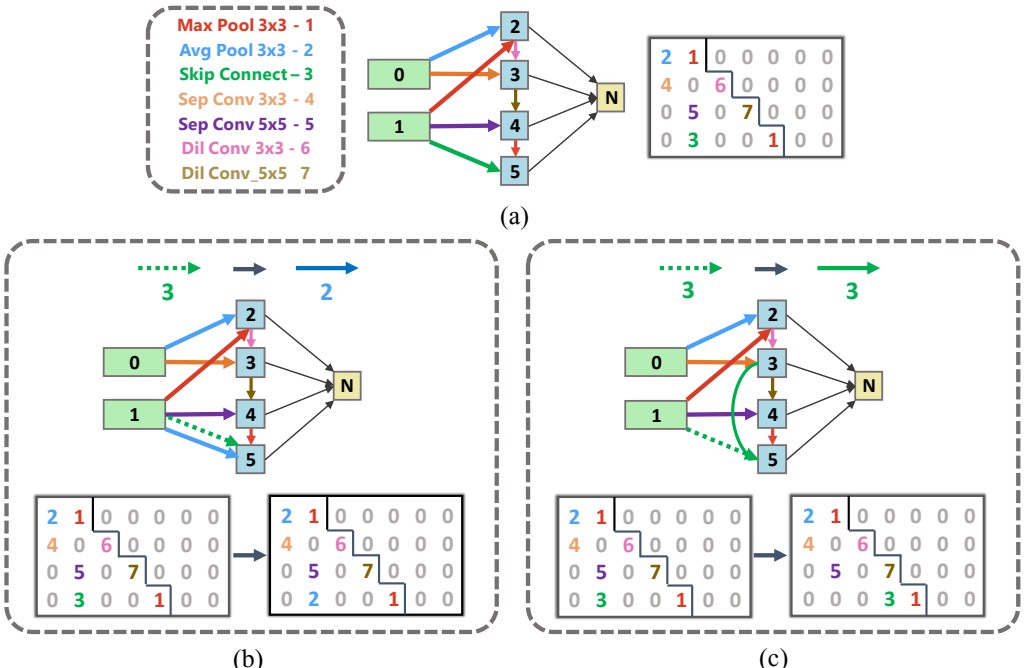

Figure 6: Matrix encoding method of PO-NAS. (a) Matrix encoding method of PO-NAS (DARTS). (b) Changes in node operation types correspond to changes in matrix elements. (c) Changes in node connection targets correspond to the swapping of matrix elements.

## A.1 Architecture Coding Method

Figure 6 illustrates the cellular matrix encoding representation of PO-NAS within the DARTS search space. With this encoding, PO-NAS can acquire structural information of specific architectures for subsequent calculations of minimal operation paths and costs. As shown in Figure 6, a unique DARTS architecture is encoded as a 4×7 matrix, where each index represents a type of connection, with specific operations depicted in the figure. The matrix can be adjusted based on the type of search space. A '0' in the matrix indicates the absence of a connection, and the matrix is segmented according to the connection patterns. In the case of the DARTS search space, since each operational node is only connected to the two preceding nodes, we allocate corresponding matrix elements based on the number of connectable nodes for each node, forming an upper triangular matrix. As DARTS includes both normal cells and reduction cells , combining two upper triangular matrices into one represents a unique DARTS structure. With this matrix representation, evolutionary searches and calculations of the shortest operation paths can be performed by simply manipulating the matrix. For

instance, operation types can be changed by altering the values of specific elements, or the connection targets of individual nodes can be modified by swapping the positions of elements within the matrix.

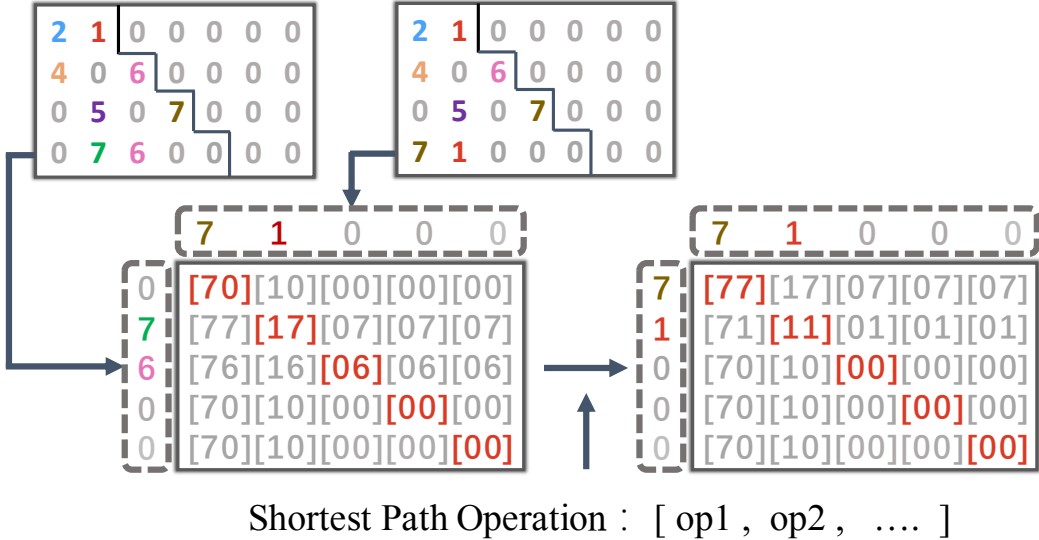

Shortest Path Operation： [ op1 , op2 , …. ]

Figure 7: Calculation method of the shortest operation path and the minimum operation cost. This method operates on a set of nodes (for example, a node vector of length 5), initially expanding two node vectors into a concatenated matrix, followed by the application of Algorithm 2 to carry out the specific computation process.

## A.2  Minimal Operation Path and Cost

Research indicates that the predictive performance differences between architectures are positively correlated with their Graph Edit Distance (GED) ((51), (52), (53), (54)). The Shortest Edit Path (SEP) effectively encodes the fundamental differences between two architectures. Inspired by these studies, we employ the shortest operation path and minimum operation cost to measure the differences between architectures. The shortest operation path is defined as the minimum set of operations required to transform one architecture into another within a set of architectures through matrix-based transformations, which include element changes and element swaps. This path not only considers the direct changes in the elements of the architecture but also the changes in the relative positions of the elements. In this way, we can more accurately assess the similarity and differences between architectures. The minimum operation cost refers to the total sum of all operations in the shortest operation path. The minimum operation cost provides a method for quantifying structural differences, allowing us to prioritize architectures during the search process that can achieve significant performance improvements with fewer operations and to measure the extent of exploration of the search space in the current search process. The specific calculation method is illustrated in Figure 7 and Algorithm 2.

Specifically, our calculations are based on a set of nodes (for example, node vectors of length 5), expanding two node vectors into a concatenated matrix. Using this concatenated matrix, we compute the shortest operation path and minimum operation cost, then sum the shortest operation paths and minimum operation costs of all nodes to obtain the overall shortest operation path and minimum operation cost between two different architectures.

In a single concatenated matrix, the elements on the diagonal represent the positional information of operations in the two node vectors, while the elements within the node vectors represent the type of operations. Our goal is to align the positional elements of the node vectors (i.e., the two elements on the diagonal are either both 0 or neither is 0) and to make the operation types of the node vectors consistent (i.e., the two elements on the diagonal are equal).

Let the individual element vector of the concatenated matrix be denoted as $A_{i,j} = (A, B)$. To minimize the operation path, we prioritize positional transformations of elements that have the same operation type but different positions: select elements off the diagonal where $A = B$, denote the column index of these elements as j, and then swap all elements B in that row with all elements B in row j (lines 2-9 of Algorithm 2). Next, we align the positional elements of the node vectors: when the number of elements on the diagonal where $A = B$ is less than 2, select elements where neither A nor B is 0, denote the row index as i and the column index as j; if element A at $A_{i,i}$ is 0 and element B at $A_{j,j}$ is also 0, swap all elements B in row i with all elements B in row j (lines 10-19 of Algorithm 2). Finally, we make the operation types of the node vectors the same: select elements on the diagonal where $A \neq B$, and set $B = A$ (lines 20-27 of Algorithm 2). We record all transformation operations and calculate the length of the operation set, thereby obtaining the shortest operation path and minimum operation cost for the architectural combination.

---

**Algorithm 2** calculation minimal operation path and cost of two nodes

---

**Require:** Concatenated matrix $\boldsymbol{A}$, the number of rows in concatenated matrix $R$
1: $C = 0, \mathbb{O} = \emptyset$
2: **for** each pair (i, j) in combinations of rows from 1 to R **do**
3:     **if** $i \neq j$ and $A_{i,j}[1] = A_{i,j}[2]$ **then**
4:         Swap every $A_{i,j}[2]_{A_{i,j} \in \boldsymbol{A}_{i,:}}$ with every $A_{i,j}[2]_{A_{i,j} \in \boldsymbol{A}_{j,:}}$
5:         Record exchange operation as $\mathcal{O}_{ex}(i, j)$
6:         $\mathbb{O} = \mathbb{O} \cup \{\mathcal{O}_{ex}(i, j)\}$
7:         $C = C + 1$
8:     **end if**
9: **end for**
10: **while** the length $L_a$ of vector $a = \{A_{i,j}, (i = j, A_{i,j}[1] \neq 0, A_{i,j}[2] \neq 0)\} < 2$ **do**
11:     **for** each pair (i, j) in combinations of rows from 1 to R **do**
12:         **if** $i \neq j$ , $A_{i,j}[1] \neq 0$, $A_{i,j}[2] \neq 0$, $A_{i,i}[1] = 0$ and $A_{j,j}[2] = 0$ **then**
13:             Swap every $A_{i,j}[2]_{A_{i,j} \in \boldsymbol{A}_{i,:}}$ with every $A_{i,j}[2]_{A_{i,j} \in \boldsymbol{A}_{j,:}}$
14:             Record exchange operation as $\mathcal{O}_{ex}(i, j)$
15:             $\mathbb{O} = \mathbb{O} \cup \{\mathcal{O}_{ex}(i, j)\}$
16:             $C = C + 1$
17:         **end if**
18:     **end for**
19: **end while**
20: **for** each pair (i, j) in combinations of rows from 1 to R **do**
21:     **if** $i = j$ and $A_{i,j}[1] \neq A_{i,j}[2]$ **then**
22:         $A_{i,j}[2] \leftarrow A_{i,j}[1]$
23:         Record change operation as $\mathcal{O}_{ch}(A_{i,j}[1], A_{i,j}[2])$
24:         $\mathbb{O} = \mathbb{O} \cup \{\mathcal{O}_{ch}(A_{i,j}[1], A_{i,j}[2])\}$
25:         $C = C + 1$
26:     **end if**
27: **end for**
28: **return** the operation set $\mathbb{O}$, the operation cost $C$

---

### A.3 Crossover and Mutation

By leveraging the low-cost evaluation advantages of the surrogate model and training-free metrics, PO-NAS is capable of focusing on and assessing a vast number of architectures without incurring significant additional costs. This approach greatly expands the search scope, allowing the algorithm to effectively explore a broad architectural space. To more efficiently guide the search process for these architectures, we introduce a novel crossover and mutation strategy. This strategy is specifically designed to simultaneously explore unknown subspaces and local optima within large search spaces.

**Shortest operation path Crossover**    PO-NAS employs a novel crossover method to generate offspring architectures. Specifically, we first calculate the shortest operation path between two parent architectures, representing the minimum set of operations required to transform one parent architecture into the other. We then randomly shuffle the operational steps in this shortest operation path to ensure diversity and exploratory nature in the search process, thereby increasing the coverage

of the search space. Subsequently, to balance the need to explore new architectural spaces with the utilization of known effective operations, we randomly select half of the shuffled operational steps for application. The chosen operations are applied to one of the parent architectures, thereby generating a new offspring architecture. Shortest operation path crossover is illustrated in Figure 8.

**Neighborhood Traversal Mutation**   After completing the crossover operation, to enhance the exploration capabilities around local optima, we traverse all neighboring architectures that have an operation cost of 1 with respect to the newly generated offspring architectures. By traversing these neighboring architectures, we are able to identify subtle changes that may lead to significant performance improvements, without straying too far from the current architecture. This approach achieves a balance between maintaining the stability of the solution and exploring new solutions. Neighborhood traversal mutation is illustrated in Figure 8.

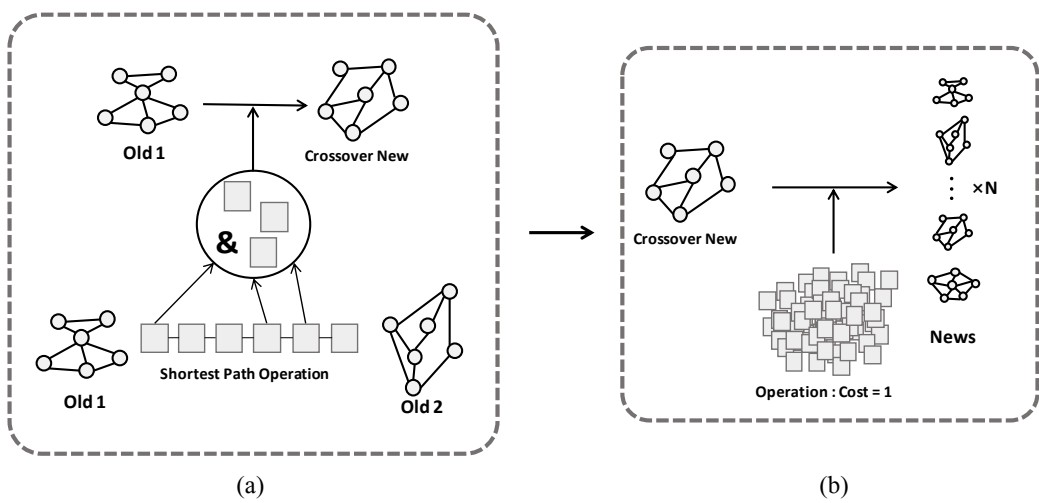

Figure 8: Shortest operation path crossover and neighborhood traversal mutation

## A.4   Search Strategy

To balance the exploration and exploitation of the search space, we introduce an exploration weight $N \in (0, 1)$. During each iteration, we select a subset of superior architectures based on the surrogate model's scores. These architectures are then paired, and the operation cost for each pair is calculated and normalized, denoted as $\hat{S}_{cost}$. Additionally, we compute the sum of surrogate model scores for these architecture pairs and normalize it, denoted as $\hat{S}_{pre}$. The final score for the architecture pair is denoted as follows:

$$\mathcal{S}_{pair} = N\hat{S}_{cost} + (1 - N)\hat{S}_{pre} \tag{9}$$

In the early stages of the search, we set a higher exploration weight $N$ to enrich the distribution of offspring architectures and enhance the exploration of the search space. As the search progresses, we reduce the exploration weight to focus more on the performance scores of the architecture pairs, aiming to identify the optimal architecture. All offspring architectures generated by the evolutionary algorithm are predicted using the surrogate model, combining the training-free metric predictor and the surrogate model weights to jointly predict scores. Given the potential errors in the training-free metric predictor, we select the top-performing offspring based on predicted scores and further calculate their actual training-free metrics to ensure accurate assessment. This approach significantly reduces the evaluation cost of offspring architectures, allowing PO-NAS to thoroughly explore large search spaces at a low training cost.

# B  Experimental Details

## B.1  Benchmark

**NAS-Bench-201**   NAS-Bench-201 (21) is a widely utilized benchmark for Neural Architecture Search (NAS), with its search space primarily composed of stacked cellular structures. Each cell consists of 4 nodes interconnected by 6 edges. The operation on each edge can be one of the following five types: 3x3 convolution, 1x1 convolution, 3x3 average pooling, zeroize, or skip connection. Consequently, the search space encompasses a total of $5^6 = 15,625$ distinct neural architectures. These architectures have been evaluated across three datasets: CIFAR-10 (C10) (55), CIFAR-100 (C100), and ImageNet-16-120 (IN-16) (56).

**TransNAS-Bench-101**   TransNAS-Bench-101 (34) offers a variety of search spaces and benchmark tasks, typically used to assess the transferability of Neural Architecture Search (NAS) algorithms across different tasks. Unlike NAS-Bench-201, which primarily focuses on cellular structures, TransNAS-Bench-101 explores both micro cellular structures and macro skeleton structures. The cellular search space is similar to that of NAS-Bench-201 but includes only four operations (excluding average pooling), resulting in a total of $4^6 = 4,096$ neural architectures. In the macro search space, the cell structure is fixed, while the skeleton is variable. The skeleton consists of 4 to 6 modules, each containing two residual blocks. Each module can choose to downsample the feature map, double the number of channels, or perform both operations. Throughout the entire architecture, downsampling may occur 1 to 4 times, and channel doubling may occur 1 to 3 times, yielding a total of 3,256 distinct architectures. TransNAS-Bench-101 evaluates these architectures on seven different visual tasks, all using a single dataset of 120K indoor scene images derived from the Taskonomy project (57). These tasks include scene classification, object detection, jigsaw puzzle solving, room layout analysis, semantic segmentation, surface normal estimation, and autoencoding.

**DARTS**   DARTS (11) is a well-known example of an extensive search space in the domain of neural architecture search (NAS). An architecture in DARTS comprises two types of cells: a normal cell and a reduction cell. Each cell features two input nodes and four intermediate nodes, where each intermediate node is connected to two randomly chosen preceding nodes via an edge. Across each edge between the nodes, a selection of seven different operations can be applied. In contrast to other benchmarks, DARTS boasts an immense search space, encompassing $10^{18}$ distinct architectures.

## B.2  Training-free Metric

**Grad_Norm**   Abdelfattah et al. (5) introduced a set of training-free scoring functions, drawing inspiration from the literature on pruning at initialization. Among these, the gradient norm (Grad_Norm) metric computes the aggregate Euclidean norm of the gradients following the propagation of a single mini-batch through the network.

**Fisher**   The Fisher metric was initially introduced in (58) to quantify the impact of model parameters on loss, facilitating the removal of activation channels with minimal loss impact through channel pruning. Turner et al. (59) aggregated the Fisher metric across all channels within a convolutional primitive to assess the significance of that primitive when substituted with a more efficient alternative. It can be formulated as follows:

$$\mathcal{S}_z(z) = \left(\frac{\partial \mathcal{L}}{\partial z} z\right)^2, \quad \mathcal{S}_n = \sum_{i=1}^{M} \mathcal{S}_z(z_i) \tag{10}$$

where $\mathcal{S}_z$ is the saliency per activation z, and $\mathcal{M}$ is the length of the vectorized feature map.

**SNIP**   Lee et al. (60) proposed performing parameter pruning based on a saliency metric computed at initialization using a single minibatch of data. This metric approximates the change in loss when a specific parameter is removed. SNIP calculates the saliency metric before training with only a single mini-batch of input data to estimate the loss variation caused by the removal of a particular parameter.

$$\mathcal{S}_p(\theta) = \left|\frac{\partial \mathcal{L}}{\partial \theta} \odot \theta\right| \tag{11}$$

where $\mathcal{L}$ is the loss function of a neural network with parameters $\theta$ and $\odot$ is the Hadamard product.

**Grasp**  Wang et al. (61) aimed to enhance the SNIP metric by approximating the gradient norm change upon parameter pruning. Grasp measures the gradient norm alteration during neural network pruning. It can be formulated as follows:

$$\mathcal{S}_p(\theta) = -\left(H\frac{\partial\mathcal{L}}{\partial\theta}\right) \odot \theta \tag{12}$$

where $H$ is the Hessian, $\mathcal{S}_p$ is the per-parameter saliency.

**SynFlow**  Tanaka et al. (62) generalized synaptic saliency scores and proposed SynFlow, a modified version that prevents layer collapse during parameter pruning. SynFlow computes a loss based on the product of all network parameters, eliminating the need for data.

$$\mathcal{S}_p(\theta) = \frac{\partial\mathcal{L}}{\partial\theta} \odot \theta \tag{13}$$

**Jacobian Covariance (Jacob_Cov)**  Jacob_Cov (63) is specifically designed to evaluate neural networks in the context of NAS. To put it simply, it reflects the correlation between activations within a neural network when influenced by different inputs in a minibatch of data. The lower the correlation, the better the network's performance, as it indicates the network's ability to effectively distinguish between various inputs.

## B.3  Implementation Details of PO-NAS

**Normalize**  To equitably assess the sensitivity of various architectures to different metrics, we normalize the training-free training metrics and metric weights using the following normalization function to map them into the interval (0,1). For the i-th training-free metric $\mathcal{Z}_i$ of a single architecture:

$$\hat{\mathcal{Z}}_i = \frac{\mathcal{Z}_i - \mathcal{Z}_{i,\min}}{\mathcal{Z}_{i,\max} - \mathcal{Z}_{i,\min}} \tag{14}$$

$\mathcal{Z}_{i,\max}$ and $\mathcal{Z}_{i,\min}$ represent the maximum and minimum values of the i-th metric across all architectures. For the i-th training-free metric weight $w_i$ of a single architecture:

$$\hat{w}_i = \frac{w_i}{\|w\|_1} \tag{15}$$

where $\|w\|_1$ denotes the L1 norm of the weight vector $w$ including the weights of all training-free metrics for a single architecture. To enable the model to account for both the positive and negative correlations of metrics, we retain the positive and negative correlations of model weights.

**Surrogate Model**  In this section, we provide a detailed description of the training setup for the surrogate model in PO-NAS. After a series of experimental tests, we identify an optimal set of training parameters and apply a uniform training strategy across all datasets. During the pre-training phase, we divide the training dataset for the architecture encoder into training and test sets in a $1 : 4$ ratio and randomly mask $20\%$ architectures for the mask reconstruction task. We train the architecture encoder using the initial architectures and their corresponding training-free metrics, employing stochastic gradient descent (SGD) over 100 epochs. In the first 5 epochs, the learning rate is initially increased to $5 \times 10^{-3}$ and then gradually reduced to 0 according to a cosine annealing schedule, with a batch size of 64. In the Bayesian Optimization (BO) phase, we set a loss threshold of 0.1 (for some tasks in TransNAS-Bench-101, adjustments are made due to the peculiarity of the loss metric values) and a maximum number of iterations of 100. Each iteration begins with a training period of 100 epochs, increasing by $10\%$ per iteration, and the model weights are reset after each iteration until the model reaches the loss threshold or the maximum number of iterations is reached, after which the best model weights are saved. The loss difference threshold $\mathcal{T}_{th}$ is set to 0.1 (adjustments are made for some tasks in TransNAS-Bench-101 due to the peculiarity of the loss metric values). We use the Adaptive Moment Estimation (Adam) optimizer to train the surrogate model. In the first 10 epochs, the learning rate is initially increased to $3 \times 10^{-4}$ and then gradually reduced to 0 according to a cosine annealing schedule, with a weight decay of 0.01. Additionally, we employ gradient clipping with max norm = 1 to prevent gradient explosion.

**PO-NAS on NAS-Bench-201 and TransNAS-Bench-101**    For NAS-Bench-201 and TransNAS-Bench-101, to assess the effectiveness of our method through ablation studies, we adopt the same training-free metrics as those used in the RoBoT experiment. These metrics include the six training-free metrics outlined by (5): grad_norm, snip, grasp, fisher, synflow, and jacob_cov. To ensure the reproducibility of the experimental results, we utilize NAS-Bench-Suite-Zero (13) to calculate these training-free metrics for both NAS-Bench-201 and TransNAS-Bench-101. Due to the relatively small search space, we utilize only the surrogate model without employing the evolutionary algorithm. For NAS-Bench-201, we maintain the experimental conditions consistent with RoBoT and HNAS, using the CIFAR-10 validation performance after 12 training epochs from the table data in NAS-Bench-201 as the objective evaluation metric for all three datasets, and calculate the search costs displayed in the same manner (i.e., the training cost of 20 architectures). However, we report the full training test accuracy of the proposed architectures after 200 epochs. For the training tasks Segmentation, Normal, and Autoencoding in TransNAS-Bench-101, to maintain consistency in experimental conditions, we do not use the training-free metric Synflow and only employ the remaining five training-free metrics. For TransNAS-Bench-101, we only report the validation performance of the architectures identified through the search process.

**PO-NAS on DARTS**    For the DARTS search space, we establish a pool of 10000 diverse architectures and calculate six training-free metrics for these architectures: grad_norm, snip, grasp, fisher, synflow, and jacob_cov. These assessments are conducted on the corresponding datasets. For the CIFAR-10 and CIFAR-100 datasets, we allocate a budget of 25 search attempts for PO-NAS, with each optimal architecture identified undergoing 10 epochs of training. For the ImageNet dataset, we set a budget of 10 search attempts, with each optimal architecture undergoing 3 epochs of training. We select three initial architectures to initialize the surrogate model, which are the top three based on the average scores of the training-free metrics. (As the surrogate model predicts architecture scores based on a combination of training-free metrics, it can still demonstrate a certain level of evaluation performance even with a limited number of initial architectures. Therefore, we do not choose to initialize more architectures.) We initiate the evolutionary algorithm at the 10th epoch (3rd epoch on ImageNet) of the Bayesian Optimization (BO) phase. Following the experimental setup of DARTS (11), we construct a 20-layer network architecture based on the identified cell structures. The initial number of channels for these architectures is set to 36, with the auxiliary tower weight set to 0.4 for CIFAR-10, located at the 13th layer; for CIFAR-100, the auxiliary tower weight is set to 0.6. We test these architectures on CIFAR-10 and CIFAR-100 through 600 epochs of stochastic gradient descent (SGD). The learning rate starts at 0.025, gradually decreasing to 0 for CIFAR-10 and from 0.035 to 0.001 for CIFAR-100, using a cosine annealing strategy. Momentum is set to 0.9, weight decay to $3 \times 10^{-4}$, and batch size to 96. Additionally, we employ Cutout (64) and ScheduledDropPath as regularization techniques, which are linearly increased from 0 to 0.2 for CIFAR-10 and from 0 to 0.3 for CIFAR-100. For ImageNet, we train a 14-layer architecture from scratch for 250 epochs with a batch size of 1024. In the first five epochs, the learning rate is initially increased to 0.7, then gradually decreases to zero according to a cosine schedule. When using the SGD optimizer, momentum is 0.9, and weight decay is $3 \times 10^{-5}$.

**Average and Best Training-free Metric Performance (Avg.) (Best)**    We report on the evaluation performance of the training-free metrics employed by PO-NAS. We compute the training-free metrics for all architectures and select the ones with the highest scores for further assessment. Specifically, "Best" refers to the architecture that demonstrated the most outstanding performance among the optimal architectures identified by separate evaluations of different metrics, while "Avg." denotes the architecture with the highest average score across all metrics.

**Search Costs of PO-NAS**    The search cost of PO-NAS is comprised of four primary components: First, the training cost of the architecture encoder during the pre-training phase; Second, the evaluation cost of real training performance during the Bayesian Optimization (BO) phase; Third, the prediction and training cost of the surrogate model in the BO phase; Lastly, the cost associated with crossover and mutation operations and the computation of offspring's training-free metrics during the evolutionary algorithm phase (this cost is omitted for NAS-Bench-201 and TransNAS-Bench-101 as they do not employ an evolutionary algorithm). For the real training performance evaluation cost of NAS-Bench-201 and TransNAS-Bench-101, we directly utilize the data provided in the benchmark tables.

## B.4  Other Implementation Details

**Classification Details of Architecture Sets in Table 5**    The architecture sets are categorized based on the proportion of operations and the magnitude of parameters. Specifically, Op0 to Op4 denote architecture sets in which the corresponding operation constitutes at least 50% of the architecture (each set comprises 1,545 architectures). In contrast, the Mix category represents architecture sets with evenly distributed operations (consisting of 1,440 architectures). For the parameter magnitude classification, each category includes 5,208 architectures. The operations are defined as follows: Op0 represents zeroize, Op1 represents skip connection, Op2 represents 1×1 convolution, Op3 represents 3×3 convolution, and Op4 represents 3×3 average pooling.

**Random Search (RS)**    We randomly sample network architectures from a predefined search space, evaluate their performance, and ultimately select the architecture with the best performance. Random search is highly flexible and makes it a common baseline for NAS algorithms.

**Regularized Evolutionary Algorithm (REA) (1)**    Regularized Evolutionary Algorithm (REA) is commonly used as a baseline for NAS algorithms. REA initializes the population by randomly selecting a batch of architectures and evaluating their performance. In our experiments, we use one-third of the current experimental budget for initialization. Then, the best-performing architecture is selected from the initial population, mutated to produce a new offspring, which is evaluated and added back to the population. For cell-based search spaces (e.g., NAS-Bench-201 and TransNAS-Bench-101-micro), we change the operation type on an edge as the mutation operation. (For TransNAS-Bench-101-macro, we randomly add or remove a downsampling/doubling operation on one residual module.)

**REINFORCE (65)**    REINFORCE is a classic policy gradient reinforcement learning algorithm that directly optimizes the policy to maximize the expected cumulative reward. In our comparative experiments, REINFORCE is used as one of the baseline algorithms. We adopt the same configuration as (21) and use the Adam optimizer with a learning rate of 0.01 to update the parameters of the policy network. Additionally, to stabilize the training process, we introduce an exponential moving average with a momentum of 0.9 as the reward baseline.

**Hybrid Neural Architecture Search (HNAS) (14)**    HNAS is one of the classic hybrid NAS algorithms, which we use as a benchmark for comparison in our experiments. HNAS employs Bayesian Optimization (BO) to optimize training-free metrics based on the true training performance of architectures. Since the training-free metrics used by HNAS are gradient-based, we follow their experimental setup and use the gradient norm as the training-free metric in our experiments.

**Robustifying and Boosting Training-free Neural Architecture Search (RoBoT) (15)**    Similar to PO-NAS, RoBoT employs a weighted linear combination to integrate multiple training-free metrics, and we use it as a benchmark for comparison in our experiments. RoBoT utilizes Bayesian Optimization (BO) to dynamically adjust the weights of these metrics based on real-time training feedback. Following their experimental setup, we use six classic training-free metrics (grad_norm, snip, grasp, fisher, synflow, and jacob_cov) as the optimized combination.

## C  More Empirical Results

### C.1  Correlation of Different Architecture Sets

We classify architectures in the NAS-Bench-201 (21) dataset based on their structural characteristics and calculate the Spearman correlation coefficients of several classic metrics across these architectural classifications. Implementation details can be found in Appendix B.4.

As shown in Table 5, the results indicate that even within the same search space, due to differences in operational composition and parameter volume, the correlation coefficients of different training-free metrics for these architectures can vary significantly. For instance, although Jacob_cov (63) generally exhibits strong performance, its correlation drops sharply to 0.33 when evaluating architectures dominated by 3×3 convolutional operations (Op3). Ignoring the differences in the assessment performance of these metrics across various types of architectures during the optimization of training-free metric combinations will inevitably affect the ability of the final metric combination to select the best architecture. Therefore, optimizing different metric combinations for different types of architectures can further enhance the evaluation capability of training-free metric combinations.

Table 5: Spearman correlation coefficients between several classic training-free metrics and the actual performance of architectures within different architecture sets on the NAS-Bench-201 dataset. Experimental details can be found in Appendix B.4

| Metrics | Operation | | | | | | Params | | | All |
|---|---|---|---|---|---|---|---|---|---|---|
| | Op0 | Op1 | Op2 | Op3 | Op4 | Mix | Low | Med | High | |
| Grad_norm | 0.74 | 0.76 | 0.06 | 0.35 | 0.71 | 0.48 | 0.55 | 0.34 | 0.22 | 0.59 |
| Snip | 0.76 | 0.75 | 0.06 | 0.36 | 0.70 | 0.48 | 0.56 | 0.33 | 0.22 | 0.60 |
| Grasp | 0.73 | 0.56 | 0.15 | 0.34 | 0.49 | 0.36 | 0.46 | 0.30 | 0.27 | 0.52 |
| Fisher | 0.69 | 0.65 | 0.02 | 0.30 | 0.69 | 0.42 | 0.46 | 0.27 | 0.17 | 0.51 |
| Synflow | 0.63 | 0.85 | 0.44 | 0.76 | 0.72 | 0.58 | 0.56 | 0.46 | 0.51 | 0.73 |
| Jacob_cov | **0.88** | **0.87** | **0.52** | 0.33 | **0.89** | 0.62 | **0.86** | **0.74** | **0.66** | **0.75** |
| zen | 0.05 | 0.86 | 0.39 | 0.60 | 0.22 | 0.50 | 0.24 | 0.09 | 0.27 | 0.35 |
| Nwo | 0.85 | 0.85 | 0.48 | 0.69 | 0.83 | **0.63** | 0.77 | 0.62 | 0.60 | **0.75** |
| Params | 0.70 | 0.42 | 0.40 | **0.79** | 0.71 | 0.58 | 0.43 | 0.42 | 0.48 | 0.72 |
| Flops | 0.64 | 0.39 | 0.32 | 0.76 | 0.69 | 0.56 | 0.35 | 0.28 | 0.42 | 0.70 |

## C.2 PO-NAS on NAS-Bench-201

We evaluate the search performance of PO-NAS on NAS-Bench-201. Due to the relatively small search space, we utilize only the surrogate model without employing the evolutionary algorithm. Table 1, Table 6, Table 7 Table 8 and Figure 9 present the performance outcomes on three benchmark datasets, with "Optimal" indicating the upper limit of this benchmark, corresponding to the architecture with the highest accuracy within the NAS-Bench-201 search space. PO-NAS demonstrates excellent performance across all three datasets. We also assess the performance of the training-free metrics we utilize, including the architectures with the highest average metric scores (Avg.) and the highest score for a single optimal metric (Best). The results show that PO-NAS significantly enhances the evaluation capability of the original metrics. As PO-NAS optimizes the metric combinations corresponding to different types of architectures, it can more effectively explore superior architectures compared to other training-free NAS methods that use global optimization metric combinations.

Table 6: Comparison of the number of architectures that must be trained for GENNAPE, $\text{FLAN}^{T}_{CAZ}$, and PO-NAS to attain the target test accuracy on NAS-Bench-201 on CIFAR - 10.

| Algorithm | **GENNAPE** (66) | $\textbf{FLAN}^{T}_{CAZ}$ (67) | **PO-NAS** |
|---|---|---|---|
| Trained models | 50 | 32 | **20** |
| Test Accuracy (%) | 93.27 | 93.30 | **94.12** |

Table 7: The mean and standard deviation of the test accuracy(%) of BOHB (68), arch2vec (69) and PO-NAS under three datasets on NAS-Bench-201. All methods search for 20 architectures. All results are reported as the mean ± standard deviation over 10 runs

| Algorithm | Accuracy (%) | | | Cost (GPU Seconds) |
|---|---|---|---|---|
| | C10 | C100 | IN-16 | |
| BOHB | 93.61±0.52 | 72.37±0.90 | 45.26±0.83 | 12000 |
| arch2vec-RL | 94.12±0.42 | 73.15±0.78 | 46.16±0.38 | 12000 |
| arch2vec-BO | **94.18±0.24** | 73.37±0.30 | 46.27±0.37 | 12000 |
| PO-NAS | 94.12±0.22 | **73.51±0.00** | **46.71±0.12** | 3162 |

## C.3 PO-NAS on TransNAS-Bench-101

To verify the robustness of PO-NAS across various training tasks, we conduct a performance evaluation on TransNAS-Bench-101. We design ablation studies to demonstrate the superior performance

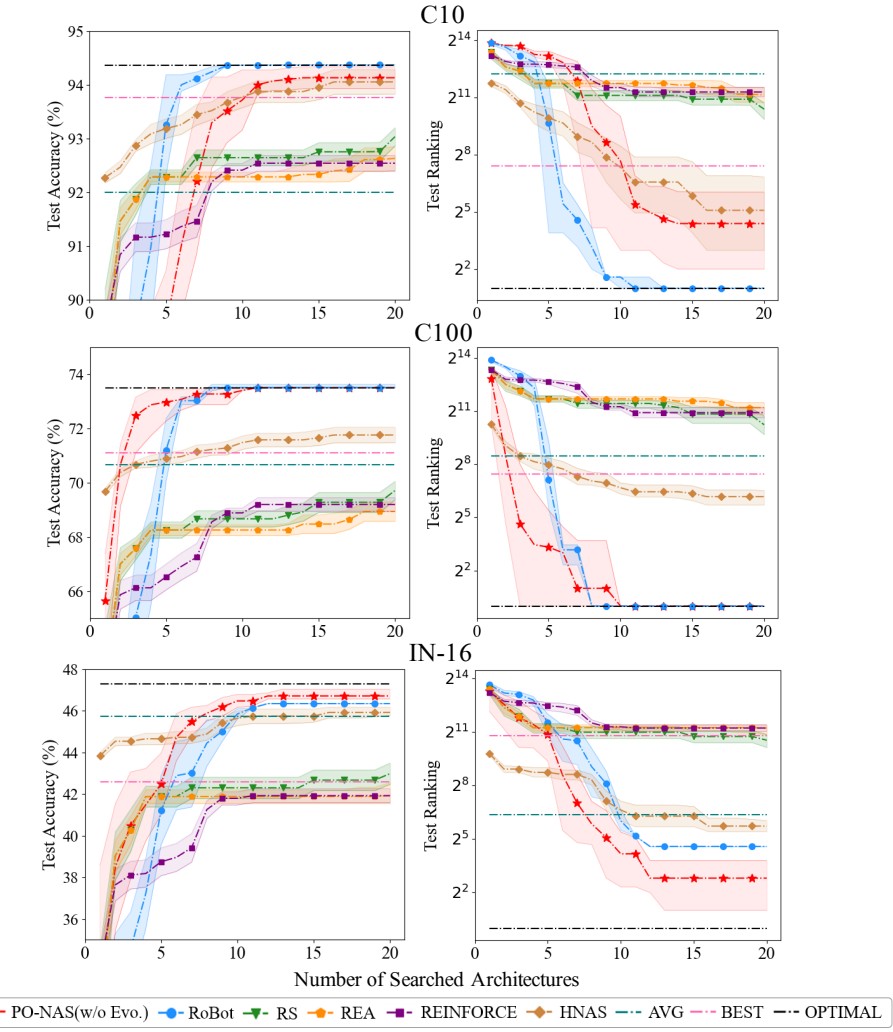

Figure 9: Comparison of various NAS algorithms in NAS-Bench-201 regarding the number of searched architectures. Results are reported with the mean and standard deviation of 10 runs, search costs are evaluated on an Nvidia 1080Ti. "AVG" and "BEST" represent the average value of metrics and the best individual metrics. The best results are in **bold**, and the second best are underlined.

of our architecture embedding-based surrogate model. Due to the relatively small search space, we utilize only the surrogate model without employing the evolutionary algorithm. We compare PO-NAS with several other hybrid NAS methods known for their robustness, as well as several benchmark search methods. Table 4, Table 9, Figure 14 and Figure 15 present the experimental results, indicating that PO-NAS achieves excellent performance across 14 different training tasks and outperforms other hybrid NAS methods in most of these tasks. This is attributed to PO-NAS's ability to optimize the corresponding metric combinations for different types of architectures. Notably, PO-NAS demonstrates significant improvements in handling the search space of macro architectures (including chain-like architectures) compared to other NAS methods and consistently identifies the optimal architectures within the search space across multiple training tasks. This confirms the great potential of PO-NAS in search spaces with regular distributions like macro architectures, as the surrogate model can more effectively optimize the metric combinations for each architecture.

Table 8: Test performance ranking comparison among various NAS algorithms on NAS-Bench-201. All methods search for 20 architectures. The results for RoBoT, HNAS, and PO-NAS are reported as the mean ± standard deviation over 10 runs, while REA, RS, and REINFORCE are evaluated over 50 runs. "Training-free (Avg./Best)" represent the average value of metrics and the best individual metrics.

| Algorithm | Test Ranking | | |
|---|---|---|---|
| | C10 | C100 | IN-16 |
| REA | 2259±550 | 2330±540 | 2399±577 |
| RS (w/o sharing) | 1346±413 | 1194±377 | 1493±363 |
| REINFORCE | 2512±400 | 1919±351 | 2400±300 |
| HNAS | 34±26 | 72±20 | 53±10 |
| RoBoT | **3±0** | **1±0** | 24±0 |
| Training-free (Avg.) | 4822 | 357 | 82 |
| Training-free (Best) | 169 | 173 | 1780 |
| PO-NAS | 21±17 | **1±0** | **7±5** |

Table 9: Test performance ranking comparison among various NAS algorithms on TransNAS-Bench-101. All methods search for 100 architectures. The results for RoBoT, HNAS, and PO-NAS are reported as the mean ± standard deviation over 10 runs, while REA, RS, and REINFORCE are evaluated over 50 runs. "Training-free (Avg./Best)" represent the average value of metrics and the best individual metrics.

| Space | Algorithm | Test Ranking | | | | | | |
|---|---|---|---|---|---|---|---|---|
| | | Scene | Object | Jigsaw | Layout | Segment. | Normal | Autoenco. |
| Micro | REA | 21±16 | 13±9 | 26±23 | 17±11 | 9±7 | 32±26 | **16±11** |
| | RS (w/o sharing) | 30±24 | 20±12 | 50±23 | 19±13 | 33±22 | 80±67 | 37±11 |
| | REINFORCE | 40±29 | 29±4 | 29±21 | 14±11 | 19±16 | 13±9 | 27±22 |
| | HNAS | 96±32 | 84±15 | 41±25 | 374±323 | 4±0 | 22±3 | 1329±56 |
| | RoBoT | 3±0 | **1±0** | 19±3 | **7±4** | 3±0 | **6±4** | 29±7 |
| | Training-free (Avg.) | 2863 | 2731 | 2701 | 2654 | 324 | 2472 | 3591 |
| | Training-free (Best) | 500 | 2372 | 1928 | 1904 | 22 | 722 | 3151 |
| | PO-NAS | **3±2** | **1±0** | 17±5 | 12±4 | **2±0** | 12±1 | 26±10 |
| Macro | REA | 9±7 | 8±5 | 26±15 | 20±12 | 10±8 | 19±14 | 18±11 |
| | RS (w/o sharing) | 14±11 | 26±22 | 31±24 | 21±13 | 32±26 | 18±15 | 20±13 |
| | REINFORCE | 27±21 | 20±16 | 13±9 | 19±12 | 36±24 | 34±27 | 31±22 |
| | HNAS | 506±40 | 535±41 | 149±27 | 50±7 | 18±0 | 202±30 | 671±31 |
| | RoBoT | 1±0 | 6±3 | 5±1 | 4±3 | 1±0 | 2±0 | 6±1 |
| | Training-free (Avg.) | 823 | 1278 | 751 | 1021 | 908 | 8 | 116 |
| | Training-free (Best) | 54 | 117 | 72 | 366 | 138 | 8 | 98 |
| | PO-NAS | **1±0** | 2±0 | 2±1 | 2±1 | **1±0** | **1±0** | 3±1 |

## C.4  PO-NAS on DARTS

We present the results of PO-NAS on DARTS in Table 2, Table 3and Table 10. DARTS cell architectures found by PO-NAS are presented in Figure 16, Figure 17 and Figure 18. With the combined efforts of the surrogate model and evolutionary algorithm, PO-NAS utilizes training-free metrics as auxiliary evaluation tools, effectively exploring the search space and demonstrating excellent performance under low-cost assessment conditions. Furthermore, PO-NAS relies solely on immediate training outcomes as supervisory signals, without the need for extensive training benchmarks, enabling it to easily migrate to large and unknown search spaces that lack substantial prior knowledge. However, PO-NAS also faces the inherent limitations of training-based NAS methods: there is still a certain gap between short-term real training performance (10 epochs/3 epochs on CIFAR-10/100/ImageNet) and the final test performance (600 epochs/250 epochs on CIFAR-10/100/ImageNet). Thus, PO-NAS is currently primarily limited to quickly searching for optimal architectures within short training periods, and bridging the gap between short-term performance

and final test performance is a goal that PO-NAS and training-based NAS methods need to further improve. Nonetheless, PO-NAS still demonstrates exceptional performance on the DARTS search space.

Table 10: Comparative results of CATE (70), arch2vec, and PO-NAS on DARTS Search Space with CIFAR-10. All results are reported as the mean ± standard deviation over 10 runs

| Algorithm | Test Error (%) | Params (M) | Search Cost |
|---|---|---|---|
| CATE-DNGO-LS (small budget) | 2.55±0.08 | 3.5 | 3.3 (GPU days) |
| CATE-DNGO-LS (large budget) | 2.46±0.05 | 4.1 | 10.3 (GPU days) |
| arch2vec-RL | 2.65±0.05 | 3.3 | 8.3 (GPU days) |
| arch2vec-BO | 2.56±0.05 | 3.6 | 9.2 (GPU days) |
| PO-NAS | 2.52±0.03 | 3.8 | 3.9 (GPU hours) |

## D    More Ablation Studies

### D.1    Ablation Study on Surrogate Model

**Ablation Study on Ensemble Method**    In this ablation study, we thoroughly investigate the impact of the surrogate model and its various components on the performance of PO-NAS. The experiments are conducted on the scene, object, jigsaw, and layout tasks in both the TransNAS-Bench-101-micro and TransNAS-Bench-101-macro datasets. We focus on three main aspects: ablation studies of global optimization versus per-architecture optimization (GP, w/o encoder), the influence of different normalization methods (w/o Negative, w/o Normal, Rank), and ablation studies of the surrogate model itself (Avg, Best). The detailed experimental settings are as follows: (a) PO-NAS: We use the default experimental settings on TransNAS-Bench-101; (b) GP: We replace the entire surrogate model with a Gaussian Process (GP) surrogate model to perform training-free metric optimization in the global search space, using the performance of the searched architectures in each round as the supervision signal to optimize the surrogate model; (c) w/o Encoder: We set the output of the architecture encoder to a same-dimensional vector with all elements equal to 1 (i.e., all architectures output the same architecture embedding), thereby eliminating the influence of the encoder and optimizing the training-free metric weights using only the attention network in the global search space; (d) w/o Negative: We eliminate the influence of weight signs, making all weight outputs positive without considering the potential negative correlation of training-free metrics; (e) w/o Normal: We directly optimize the numerical values of the training-free metrics without normalizing them; (f) Rank: Instead of using the numerical values of the training-free metrics directly, we use the ranking of a single architecture among all architectures for a particular training-free metric as the training-free metric; (g) Avg: We eliminate the surrogate model and select architectures using the average value of all training-free metrics; (h) Best: We eliminate the surrogate model and choose the optimal architecture from those selected by individual training-free metrics as the result; (i) OPTIMAL: The performance of the optimal architecture on the entire dataset.

Figure 10 presents our experimental results, which fully demonstrate the effectiveness of our surrogate model and its various components. PO-NAS achieves a higher performance ceiling in the global search space by assigning training-free metric weights to each specific architecture, compared to the other two methods. As the search progresses, PO-NAS is able to discover architectures with superior performance. Moreover, distinguishing between the positive and negative correlations of training-free metrics allows PO-NAS to leverage potentially negative-correlated metrics to enhance performance. Normalizing the metrics can effectively reduce the optimization difficulty of the surrogate model. However, since ranking metrics forces uniformity in metric value differences, while the actual differences in training-free metric values among architectures with similar rankings may vary, using metric ranking as a proxy metric may hinder the surrogate model's ability to accurately measure the performance of different architectures. Nevertheless, in certain tasks (such as macro-jigsaw), the Rank method exhibits unexpected performance. Therefore, in practical applications, we recommend experimenting with different combinations based on the specific training task, which may lead to unexpected performance improvements.

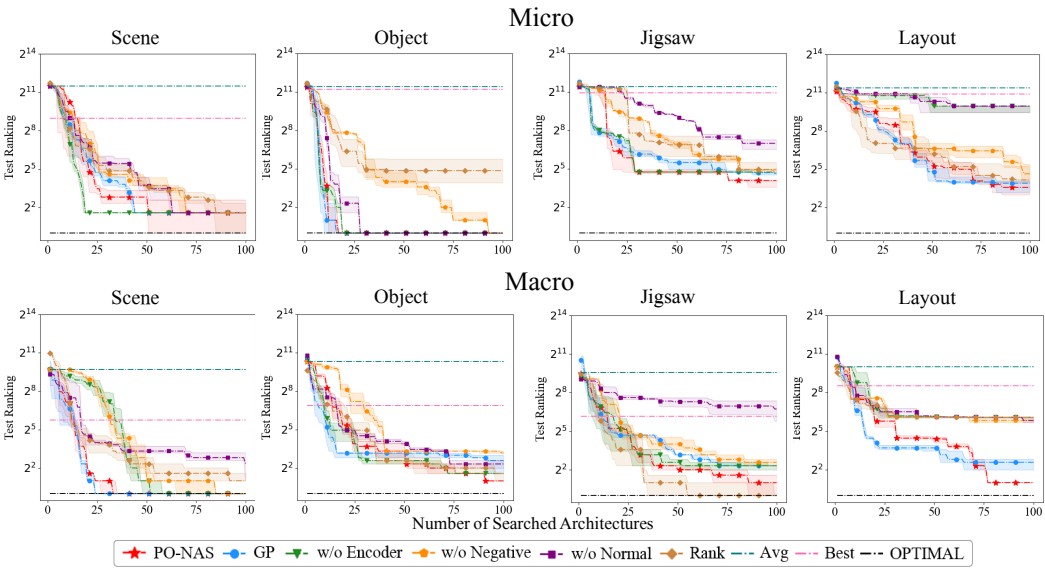

Figure 10: Comparison between different ensemble methods on 8 tasks in TransNAS-Bench-101 regarding the number of searched architectures. Note that all methods are reported with the mean and standard error of 10 independent searches.

**Ablation Study on Pre-training Loss** To delve into the performance of the architecture encoder and the training-free metric predictor, we calculate the predicted loss of various training-free metrics for scene, object, jigsaw, and layout tasks in TransNAS-Bench-101-micro and TransNAS-Bench-101-macro. The results, as shown in Table 11 and Figure 11, indicate that across different training tasks, the predicted loss of the metric predictor is concentrated between $0.5 \times 10^{-3}$ and $1.5 \times 10^{-3}$, which corresponds to an error margin of $2\%$ to $3\%$. This demonstrates that our metric predictor possesses a certain level of predictive capability. Therefore, to prevent the impact of prediction errors on the selection of top-level architectures, PO-NAS utilizes the metric predictor for pre-scoring during the evolutionary algorithm phase. It selects outstanding offspring from the pre-scoring and then recalculates the metrics to obtain the final scores. Additionally, in our experiments, we find that the predicted loss of training-free metrics also varies significantly across different search spaces and prediction tasks. For instance, Synflow exhibits a considerably larger predicted loss in the micro space compared to other metrics but has a very small predicted loss in the macro space. The predicted loss to some extent reflects the connection between training-free metrics and architectural features, which also affects the predictive accuracy of the subsequent surrogate model in the process. This experiment provides us with insights: For different search spaces and training tasks, we should selectively choose different combinations of metrics (with smaller predicted losses), rather than using a fixed single combination of metrics, which may enhance the predictive performance of PO-NAS. We conduct a rigorous analysis of the architectural features captured by each metric and employ the sensitivity analysis methods from DARTS-IM (71) to quantitatively assess the complementarity of the metrics.

**Ablation Study on Architecture Encoder Supervision Signals** We investigate the impact of different architecture encoder supervision signals on the performance of PO-NAS. As a comparison, we employed recently developed architectural encoders that use actual training as supervision signals. Among them, the experimental setup of arch2Vec on NAS-Bench-201 is most comparable to PO-NAS. To preserve the original performance of arch2Vec, we retained its official training conditions: obtaining its fixed embeddings through supervision by actual architecture performance, and then feeding these embeddings into the attention network for architecture discrimination, followed by a direct comparison with PO-NAS. The preliminary results are summarized in Table 12. The results indicate that replacing PO-NAS's encoder with existing SOTA architectural encoders yields only marginal improvements on certain datasets, while the overall performance remains virtually unchanged. This further corroborates the excellence of PO-NAS's encoder and demonstrates that

Table 11: Test loss of the metric predictor for various metrics on 8 tasks in TransNAS-Bench-101 with a 1:4 training-to-testing set ratio over 100 epochs. The results are reported as the mean ± standard deviation over 10 runs.

| Space | Tasks | Mse Loss ($\times 10^{-3}$) | | | | | | |
|---|---|---|---|---|---|---|---|---|
| | | Grad_norm | Snip | Grasp | Fisher | Synflow | Jabcob_cov | Total |
| Micro | Scene | 0.58±0.05 | 0.67±0.05 | 0.63±0.03 | 0.16±0.04 | 1.48±0.38 | 0.27±0.07 | 0.63±0.10 |
| | Object | 0.67±0.12 | 0.60±0.91 | 0.91±0.01 | 0.17±0.04 | 1.85±0.21 | 1.83±1.64 | 1.31±0.49 |
| | Jigsaw | 0.47±0.01 | 0.67±0.02 | 0.09±0.00 | 0.10±0.01 | 1.46±0.27 | 0.38±0.17 | 0.53±0.08 |
| | Layout | 0.86±0.02 | 0.78±0.01 | 0.33±0.03 | 0.20±0.01 | 1.47±0.17 | 0.28±0.16 | 0.65±0.07 |
| Macro | Scene | 1.45±0.07 | 0.74±0.05 | 2.94±0.03 | 1.44±0.08 | 0.10±0.03 | 2.00±0.09 | 1.45±0.06 |
| | Object | 1.12±0.08 | 0.70±0.07 | 0.72±0.02 | 1.48±0.13 | 0.10±0.02 | 1.08±0.03 | 0.87±0.06 |
| | Jigsaw | 0.39±0.08 | 0.32±0.07 | 0.82±0.01 | 0.69±0.07 | 0.14±0.03 | 1.34±0.05 | 0.62±0.05 |
| | Layout | 2.01±0.05 | 1.22±0.03 | 1.73±0.02 | 1.41±0.01 | 0.11±0.03 | 1.64±0.11 | 1.35±0.04 |

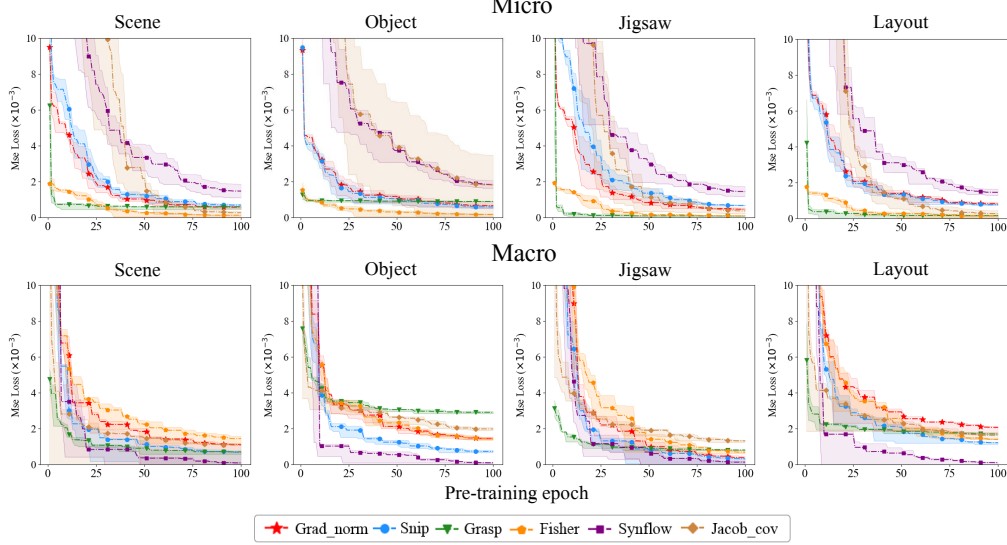

Figure 11: Test loss of the metric predictor for various metrics on 8 tasks in TransNAS-Bench-101 with a 1:4 training-to-testing set ratio over 100 epochs. The results are reported as the mean ± standard deviation over 10 runs.

learning architecture representations via training-free metrics is not only feasible but also highly efficient, significantly reducing the cost of acquiring topological information.

Table 12: Impact of Different Encoders Supervisory Signals on PO-NAS (reported by test accuracy (%) on NAS-Bench-201). All results are reported as the mean ± standard deviation over 10 runs

| Encoder | CIFAR-10 | CIFAR-100 | ImageNet-16-120 | Supervisory |
|---|---|---|---|---|
| arch2vec | 94.25±0.18 | 73.51±0.00 | 46.49±0.37 | short-term training (10 epochs) |
| PO-NAS | 94.12±0.22 | 73.51±0.00 | 46.71±0.12 | train-free metrics |

**Ablation Study on Difference Threshold $\mathcal{T}_{\text{th}}$ and Loss Threshold**   To investigate the impact of the difference threshold $\mathcal{T}_{\text{th}}$ (Section 3.4) and the loss threshold (Appendix B.3) on the prediction performance of the surrogate model, we compare the search performance of different parameter settings (with the difference threshold on the x-axis and the loss threshold on the y-axis) across the scene, object, jigsaw, and layout tasks in TransNAS-Bench-101-micro and TransNAS-Bench-101-macro. Table 15 presents our experimental results. The results demonstrate that different combinations of thresholds significantly influence the prediction performance of the surrogate model.

Lower combinations of loss and difference thresholds generally enhance the model's prediction accuracy but slow down the training speed. A higher difference threshold allows the surrogate model to more easily reach a lower loss threshold but increases the number of ranking errors, thereby affecting the final results. Moreover, a higher loss threshold, regardless of the difference threshold setting, consistently degrades the model's prediction accuracy. Therefore, to balance model fitting speed and prediction accuracy, we recommend setting lower loss and difference thresholds while maintaining a certain fitting speed, and ensuring that the gap between the two thresholds is not too large.

Table 13: Comparison of surrogate models with different parameters (X-axis: Difference Threshold, Y-axis: Loss Threshold) on TransNAS-Bench-101 across 8 tasks regarding the number of searched architectures. All models search for 100 architecture. The results are reported as the mean ± standard deviation over 10 runs.

| Space | Loss\Differ. ($\times 10^{-1}$) | Test Rank | | | | | | | |
|---|---|---|---|---|---|---|---|---|---|
| | | 1 | 2 | 3 | 4 | 1 | 2 | 3 | 4 |
| | | Scene | | | | Object | | | |
| | 1 | 3±2 | 3±0 | 3±0 | 4±3 | 1±0 | 1±0 | 1±0 | 1±0 |
| | 2 | 3±2 | 3±2 | 3±2 | 6±3 | 1±0 | 1±0 | 1±0 | 1±0 |
| | 3 | 5±3 | 5±3 | 7±5 | 7±5 | 1±0 | 1±0 | 1±0 | 1±0 |
| | 4 | 6±3 | 5±3 | 6±4 | 7±5 | 1±0 | 1±0 | 1±0 | 1±0 |
| Micro | | Jigsaw | | | | Layout | | | |
| | 1 | 17±5 | 17±5 | 17±5 | 17±5 | 12±4 | 12±4 | 19±11 | 12±4 |
| | 2 | 19±3 | 17±5 | 19±3 | 17±5 | 12±4 | 19±11 | 12±4 | 21±8 |
| | 3 | 19±0 | 19±0 | 19±3 | 19±3 | 19±11 | 17±5 | 29±12 | 19±11 |
| | 4 | 19±3 | 17±5 | 19±0 | 19±3 | 112±81 | 51±32 | 29±12 | 37±16 |
| | | Scene | | | | Object | | | |
| | 1 | 1±0 | 1±0 | 1±0 | 1±0 | 2±0 | 2±0 | 2±0 | 3±0 |
| | 2 | 1±0 | 1±0 | 1±0 | 1±0 | 2±0 | 2±0 | 3±0 | 2±0 |
| | 3 | 1±0 | 1±0 | 1±0 | 1±0 | 2±0 | 2±0 | 2±0 | 3±0 |
| | 4 | 1±0 | 1±0 | 1±0 | 1±0 | 2±0 | 2±0 | 3±0 | 3±0 |
| Macro | | Jigsaw | | | | Layout | | | |
| | 1 | 2±1 | 2±1 | 3±1 | 3±2 | 2±1 | 2±0 | 2±0 | 3±2 |
| | 2 | 2±1 | 2±1 | 2±1 | 3±1 | 2±0 | 2±1 | 2±0 | 3±2 |
| | 3 | 3±2 | 2±1 | 2±1 | 2±1 | 2±0 | 2±1 | 2±0 | 2±0 |
| | 4 | 3±1 | 2±1 | 3±2 | 3±2 | 2±0 | 3±2 | 2±0 | 3±2 |

**Ablation Study on Number of Training-free Metrics** In this ablation study, we investigate the impact of the number of training-free metrics on the prediction performance of the surrogate model. We compare the search performance of surrogate models with different numbers of training-free metrics across the scene, object, jigsaw, and layout tasks in TransNAS-Bench-101-micro and TransNAS-Bench-101-macro. The specific settings are as follows: (a) 6 Metrics: Consistent with the original setting of PO-NAS (Appendix B.3), we use six metrics: grad_norm, snip, grasp, fisher, synflow, and jacob_cov; (b) 8 Metrics: We add two additional metrics (params and flops) to the original six metrics; (c) 4 Metrics: We remove two metrics with lower relevance (grasp and fisher) from the original six metrics; Figure 12 presents our experimental results. The findings indicate that when more training-free metrics are used, the need to assign different metric weights for each architecture requires more training samples for surrogate model training. This leads to reduced stability and prediction performance in the early and middle stages of the search. However, as more architectures are evaluated during the search, the higher number of metrics provides a greater upper limit on performance. Conversely, when fewer training-free metrics are used, the surrogate model quickly achieves a certain level of prediction performance and identifies promising architectures in the early stages of the search. However, in later stages, especially for more complex tasks, its performance drops significantly compared to other methods. Therefore, in practical applications, the number of training-free metrics should be chosen based on the complexity of the training task and the search cost budget. Given the fitting challenges of the surrogate model, improving the quality

of metrics (e.g., replacing existing metrics with more relevant ones) is more effective for enhancing performance than simply increasing the number of metrics.

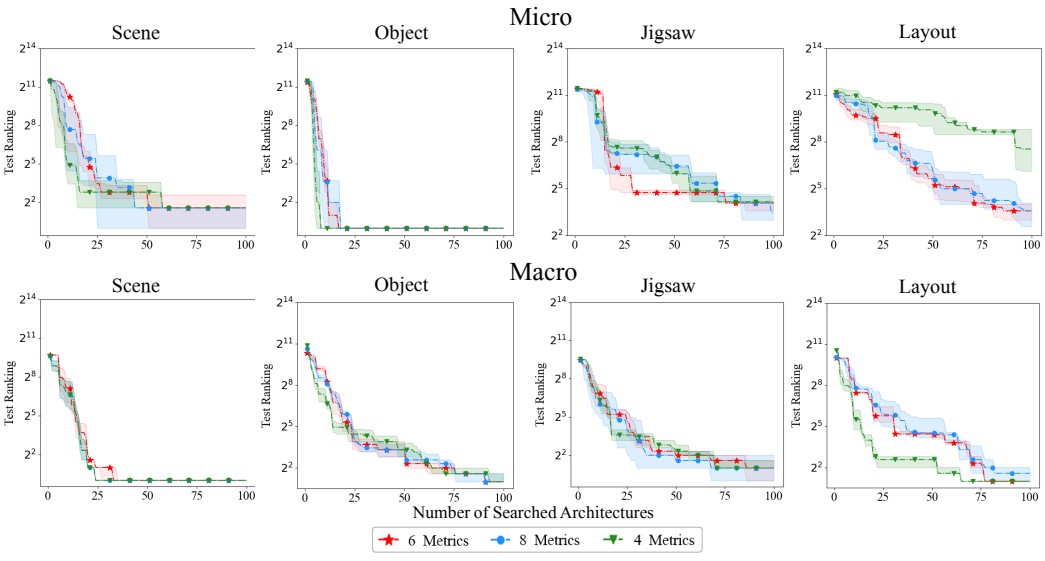

Figure 12: Comparison between utilizing different numbers of training-free metrics on 8 tasks in TransNAS-Bench-101 regarding the number of searched architectures. Note that all methods are reported with the mean and standard error of 10 independent searches.

### D.2 Ablation Study on Evolutionary Algorithm

In this ablation study, we thoroughly investigate the impact of the proposed evolutionary algorithm and its various components on the performance of PO-NAS. Specifically, we conduct 100 search cycles in the DARTS search space, using our surrogate model in combination with different evolutionary algorithm designs. We evaluate their performance based on the test accuracy after 10 training epochs for comparison. To fully demonstrate the ability of our evolutionary algorithm to explore and supplement the initial architecture population, we conduct experiments with different initial population sizes (100, 1000, 10000, and 60000). To avoid the impact of initial population size on the training performance of the architecture encoder during the pre-training phase, the same architecture encoder (trained with 10000 architectures) is used for all experiments with different initial population sizes. The evolutionary algorithm is applied in all 100 search cycles, and we ablate different components of the evolutionary algorithm while maintaining relatively fair experimental conditions, as follows: (a) PO-NAS: In each search cycle, we select the top 20 combinations for crossover operations and perform neighborhood mutation operations (generating 128 offspring), resulting in a total of 2560 offspring. Finally, we select the top 50 offspring to supplement the initial population; (b) Without Evolutionary Algorithm (w/o Evolution): In this approach, we do not use the evolutionary algorithm and rely solely on the surrogate model for prediction. All architectures are derived from the initial population; (c) Without Exploration Weight $N$ (w/o N): In this method, we ablate the exploration weight $N$. All parent combinations are selected based solely on the surrogate model's scores, ignoring the impact of the architectural combination operation cost; (d) Without Neighborhood Mutation (w/o Mutation): In this method, we ablate the mutation operation. All offspring are generated through shortest-path crossover operations. To maintain fair experimental conditions, we select the top 20 combinations in each cycle, with each combination generating 128 offspring, resulting in a total of 2560 offspring. Finally, we select the top 50 offspring to supplement the initial population; (e) Without Shortest-Path Crossover (w/o Crossover): In this method, we ablate the crossover operation. All offspring are generated through neighborhood mutation operations. To maintain fair experimental conditions, we select the top 20 architectures in each cycle, with each architecture generating 128 offspring through neighborhood mutation, resulting in a total of 2560 offspring. Finally, we select the top 50 offspring to supplement the initial population; (f) Regularized Evolutionary Algorithm (REA) (1): We use REA as a benchmark for the evolutionary algorithm.

Following its experimental settings, we select the best architecture for mutation operations in each cycle. We define the mutation operation as randomly changing the operation type and edge object of one operation edge. Like the other methods, we generate 2560 offspring in each cycle and select the top 50 offspring to supplement the initial population.

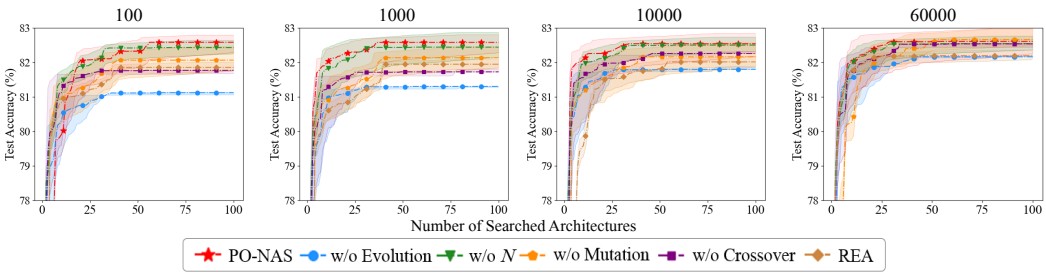

Figure 13: Comparison between different evolution methods on CIFAR-10 on DARTS regarding the number of searched architectures. Note that all methods are reported as the 10-epochs test accuracy with the mean and standard error of 10 independent searches.

Figure 13 presents our experimental results. When the initial population size is relatively small, all evolutionary algorithms show significant performance improvements compared to the method without an evolutionary algorithm. Among them, the evolutionary algorithm with crossover operations performs particularly well, outperforming methods that generate offspring solely through mutation operations. This highlights the potential of crossover operations in exploring unknown search spaces. The performance of PO-NAS is very close to that of the method without the exploration weight $N$. Although the exploration weight $N$ sacrifices some stability, it achieves a higher search upper limit. When the initial population size reaches a certain scale, the performance gap between different methods further narrows. At this point, the initial population is sufficiently diverse, covering a wide range of architectures. Therefore, exploring the local area through mutation operations alone can still yield good results. Among them, the method with only crossover operations (w/o Mutation) achieves the highest performance upper limit compared to other methods. At this time, PO-NAS does not show a clear advantage. Nevertheless, the complete PO-NAS evolutionary algorithm consistently achieves stable and excellent performance across different initial population sizes, without being limited by the initial population size.

We conducted experiments to investigate the impact of the evolution starting cycle ($T_e$) and the number of parents ($N_{par}$) and offspring ($N_{off}$) per cycle on the performance of the evolution method. The results in Tables 14 and Tables 15 demonstrate a clear negative correlation between $T_e$ and final performance, as accuracy consistently declines when $T_e$ exceeds 10. This indicates the necessity of reserving a certain search budget for the evolution method to ensure adequate exploration. In terms of population size, a larger number of $N_{off}$ exhibits a weakly positive correlation with accuracy, suggesting that increasing offspring diversity is beneficial. In contrast, $N_{par}$ shows a less systematic influence. Thus, the optimal strategy should balance the allocation of $N_{off}$ and $T_e$ within the given computational budget.

Table 14: Comparison between different $\mathbf{T_e}$ of evolution method on CIFAR-10/CIFAR-100 on DARTS. Note that all methods are reported as the 10-epochs test accuracy with the mean and standard error of 10 independent searches.

| DateSet\$\mathbf{T_e}$ | Accuracy (%) | | |
|---|---|---|---|
| | **0** | **5** | **10** |
| CIFAR-10 | 82.69±0.41 | 82.71±0.39 | 82.63±0.33 |
| CIFAR-100 | 61.31±0.28 | 61.26±0.31 | 61.19±0.27 |
| | **15** | **20** | **25** |
| CIFAR-10 | 82.48±0.42 | 82.23±0.28 | 81.84±0.11 |
| CIFAR-100 | 60.87±0.44 | 60.36±0.57 | 59.27±0.37 |

Table 15: Comparison of evolution method with different parameters (X-axis: Number of offspring per epoch ($\mathbf{N_{off}}$), Y-axis: Number of parent per epoch ($\mathbf{N_{par}}$)) on on CIFAR-10/CIFAR-100 on DARTS. Note that all methods are reported as the 10-epochs test accuracy with the mean and standard error of 10 independent searches.

| $\mathbf{N_{par}}$\$\mathbf{N_{off}}$ | Accuracy (%) | | | |
|---|---|---|---|---|
| | 640 | 1280 | 2560 | 5120 |
| CIFAR-10 | | | | |
| 5 | 82.36±0.28 | 82.43±0.36 | 82.47±0.36 | 82.43±0.41 |
| 10 | 82.41±0.35 | 82.48±0.29 | 82.52±0.33 | 82.61±0.26 |
| 20 | 82.39±0.41 | 82.40±0.37 | 82.59±0.26 | 82.66±0.32 |
| 40 | 82.46±0.36 | 82.51±0.21 | 82.65±0.32 | 82.64±0.22 |
| CIFAR-100 | | | | |
| 5 | 61.09±0.21 | 61.08±0.36 | 61.21±0.39 | 61.14±0.40 |
| 10 | 61.00±0.41 | 61.08±0.50 | 61.17±0.29 | 61.23±0.30 |
| 20 | 61.07±0.12 | 61.19±0.24 | 61.11±0.31 | 61.26±0.26 |
| 40 | 61.16±0.33 | 61.13±0.22 | 61.23±0.24 | 61.19±0.32 |

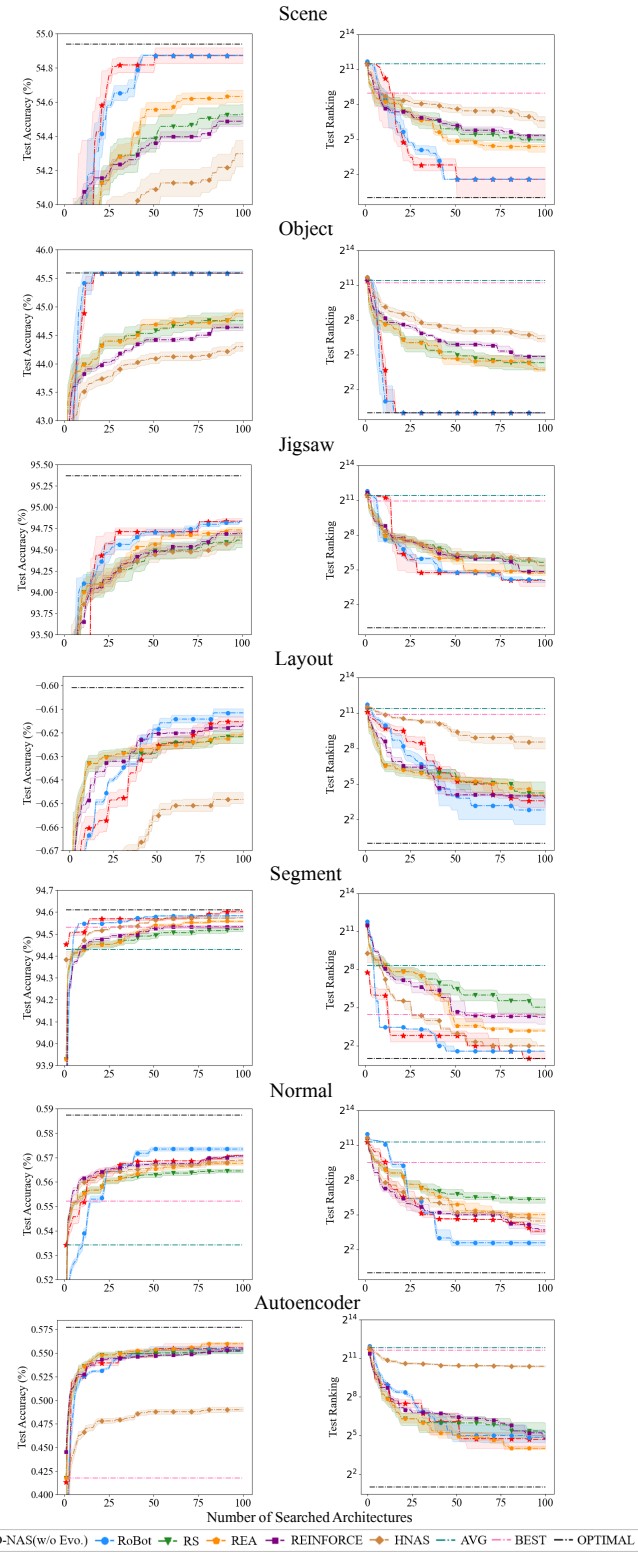

Figure 14: Comparison of various NAS algorithms in TransNAS-Bench-101-micro regarding the number of searched architectures. The results for RoBoT, HNAS, and PO-NAS are reported as the mean and standard deviation over 10 runs, while REA, RS, and REINFORCE are evaluated over 50 runs. "AVG" and "BEST" represent the average value of metrics and the best individual metrics.

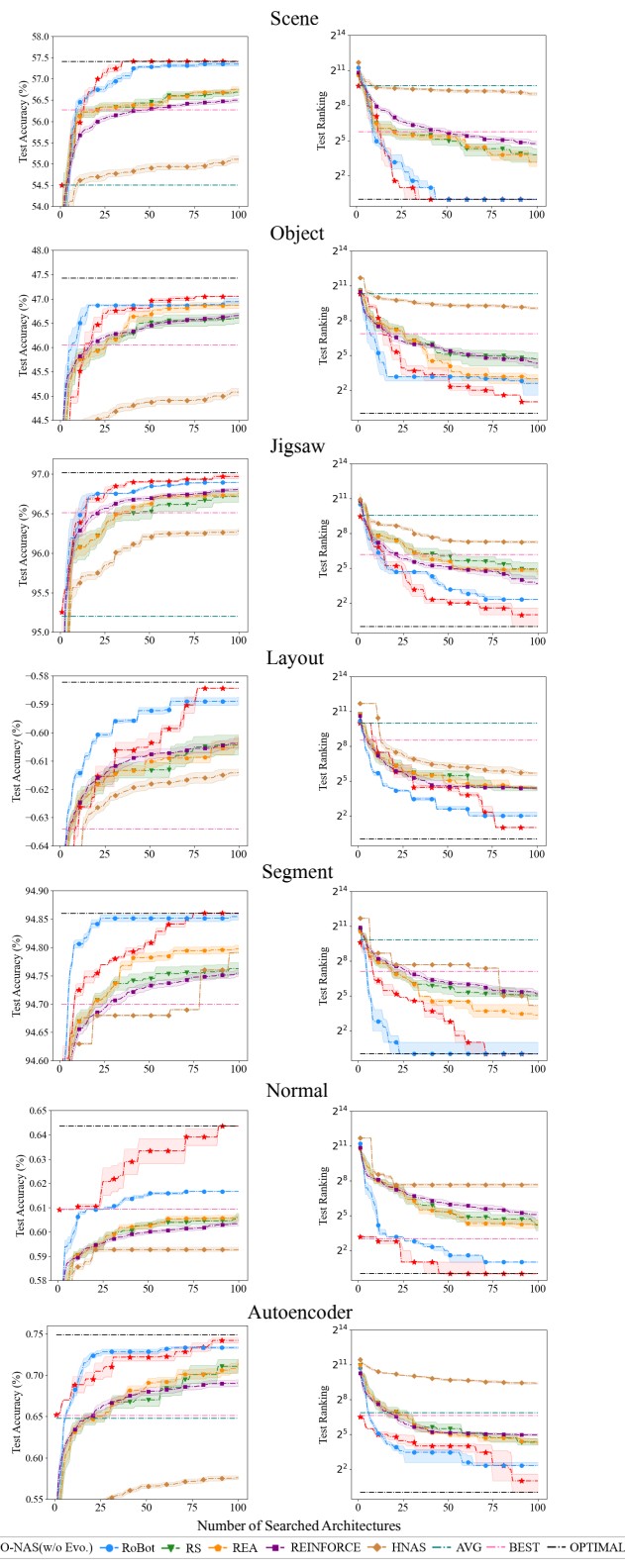

Figure 15: Comparison of various NAS algorithms in TransNAS-Bench-101-macro regarding the number of searched architectures. The results for RoBoT, HNAS, and PO-NAS are reported as the mean and standard deviation over 10 runs, while REA, RS, and REINFORCE are evaluated over 50 runs. "AVG" and "BEST" represent the average value of metrics and the best individual metrics.

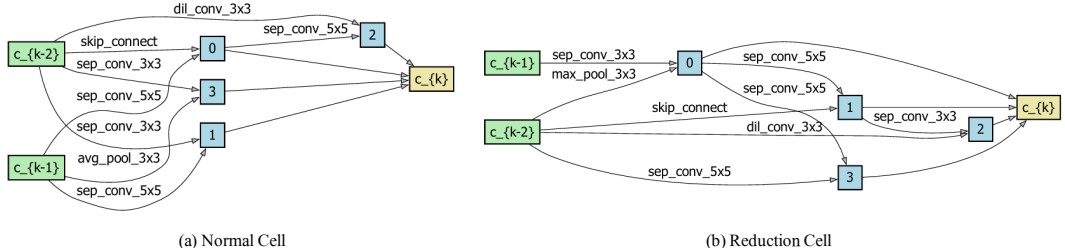

Figure 16: DARTS cell architecture found by PO-NAS on CIFAR-10 dataset with model size 3.85 MB.

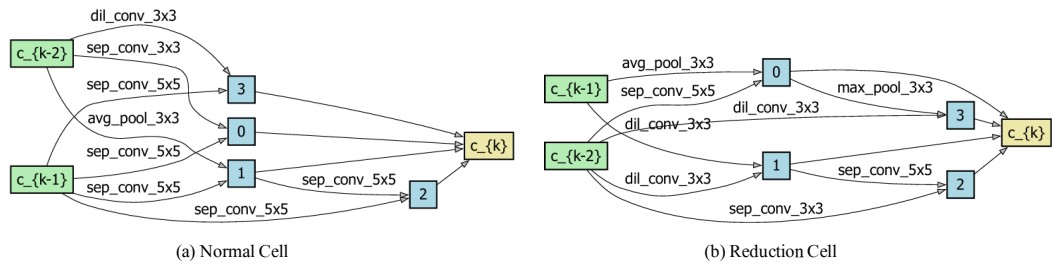

Figure 17: DARTS cell architecture found by PO-NAS on CIFAR-100 dataset with model size 4.22 MB.

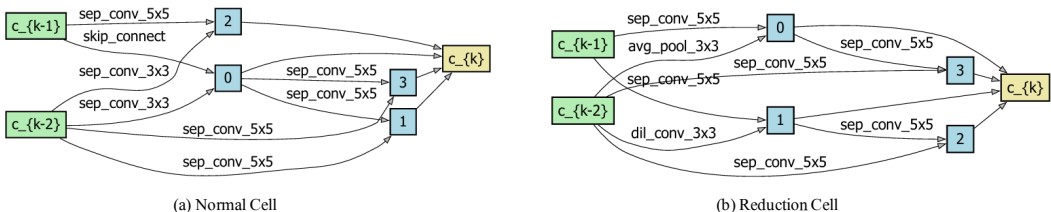

Figure 18: DARTS cell architecture found by PO-NAS on ImageNet dataset with model size 6.25 MB.

