# OpenReview forum: "Per-Architecture Training-Free Metric Optimization for Neural Architecture Search"
_NeurIPS.cc/2025/Conference — NeurIPS 2025 poster_

### Official Review · Reviewer_KFhC · 2025-06-23

**Clarity:** 2
**Significance:** 3
**Originality:** 3
**Rating:** 4
**Confidence:** 4

**Summary:**

The authors propose a novel method called PO-NAS, which integrates training-free metrics and individually optimizes metric combinations for each architecture. The approach significantly enhances search efficiency and performance by incorporating evolutionary algorithms.

**Questions:**

1. Do multiple runs with different seeds yield different architectures? Approximately how many attempts are needed to obtain the optimal architecture's best results?
2. Could the authors provide either intuitive explanations or rigorous theoretical proofs to demonstrate how these metrics contribute to the search process?

**Ethical Concerns:**

["NO or VERY MINOR ethics concerns only"]

**Final Justification:**

please see response

**Limitations:**

yes

**Quality:**

2

**Strengths And Weaknesses:**

Strengths:
1. The authors integrate multiple metrics and optimize their combinations through limited training to obtain relatively effective weighting schemes.
2. They employ evolutionary algorithms combined with trained metrics predictors to extensively explore the search space.
3. The superiority and scalability of PO-NAS are validated across multiple datasets.

Weaknesses:
1. The authors fail to explain the specific meanings of each metric in Table 4, making certain details of the paper difficult to comprehend.
2. While the core contribution focuses on utilizing multiple metrics and searching for optimal weightings through training, there's no discussion about why these metrics make the architecture search effective. This fundamentally limits its extensibility. The authors should instead explore the interpretability of these metrics (similar to works like DARTS-IM [1]), which would be more meaningful for NAS development.

[1].Zhang M, Huang W, Yang B. Interpreting operation selection in differentiable architecture search: A perspective from influence-directed explanations[J]. Advances in Neural Information Processing Systems, 2022, 35: 31902-31914.

---

> ### Author Rebuttal · Authors · 2025-07-29
>
> We deeply value your thorough assessment and insightful feedback, which precisely target the key facets of our method; the corresponding clarifications are provided in the succeeding sections.
>
> **Weaknesses**
>
> > The authors fail to explain the specific meanings of each metric in Table 4, making certain details of the paper difficult to comprehend.
>
> We sincerely apologize for not providing detailed explanations of each metric's specific meaning in Table 4, and we truly appreciate you highlighting this issue which is indeed crucial for our paper's readability. In response to your valuable feedback, we will supplement the fundamental definitions of each training-free metric along with relevant literature references in the appendix of the revised manuscript.
>
> Additionally, regarding the specific implementation methods for operation subset classification, we have already provided detailed explanations in Appendix B.3 and have added corresponding appendix index markers in the table title. We believe these supplementary explanations will effectively enhance the paper's readability.
>
> > While the core contribution focuses on utilizing multiple metrics and searching for optimal weightings through training, there's no discussion about why these metrics make the architecture search effective. This fundamentally limits its extensibility. The authors should instead explore the interpretability of these metrics (similar to works like DARTS-IM [1]), which would be more meaningful for NAS development.
>
> We appreciate your insightful comments, which have accurately identified a critical theoretical gap in our work. While PO-NAS has demonstrated promising results through dynamic combinations of multiple training-free metrics, we acknowledge the need to strengthen the theoretical foundation explaining why these particular metrics are effective.
>
> We fully recognize the value of the theoretically-driven analysis presented in DARTS-IM [1], particularly their seminal findings regarding the relationship between second-order information and architectural performance. Additionally, we found the sensitivity analysis framework of DARTS-IM to be indispensable for understanding how training-free metrics influence the ultimate performance of PO-NAS. It will help readers better understand our approach. Consequently, in the revised manuscript, we will systematically elaborate on the theoretical principles underlying each employed training-free metric. This will include a rigorous analysis of the architectural characteristics captured by individual metrics and the application of sensitivity analysis methods from DARTS-IM to quantitatively assess metric complementarity. These enhancements will substantially improve the theoretical depth and extensibility of PO-NAS, while appropriately acknowledging the foundational theoretical contributions of DARTS-IM through comprehensive citations at relevant junctures.
>
> **Questions**
>
> > Do multiple runs with different seeds yield different architectures? Approximately how many attempts are needed to obtain the optimal architecture's best results?
>
> We appreciate your concern regarding the stability of PO-NAS, especially the potential for performance variability across different task settings.
>
> To fully address this, every experiment on TransNAS-Bench-101 and NAS-Bench-201 is repeated with multiple independent runs, each initialized with a distinct random seed (for instance, seeds 0–9 are cycled whenever ten or more replications are performed), and results are reported as mean ± standard deviation. In several search tasks, PO-NAS consistently identifies the theoretically optimal architecture, i.e., identical structures are recovered across runs. In larger search spaces such as DARTS, different seeds can yield non-identical architectures accompanied by minor performance fluctuations; nevertheless, the architectures discovered by PO-NAS retain competitive and stable performance across all evaluated datasets.
>
> > Could the authors provide either intuitive explanations or rigorous theoretical proofs to demonstrate how these metrics contribute to the search process?
>
> Thank you for your incisive observation. We apologize for not explicitly clarifying the adopted metrics in the manuscript.
>
> It should be emphasized that the core contribution of PO-NAS lies in the dynamic composition mechanism of training-free metrics and the investigation of their generalization capability, rather than in the theoretical novelty of any
>  single metric. The interpretability of each metric and its mechanism for facilitating the NAS search process have already been rigorously proven in the foundational literature [2, 3, 4, 5].
>
> We fully agree on the necessity of strengthening theoretical exposition; therefore, we will supplement the revised manuscript with detailed descriptions of the precise meaning and theoretical basis of every training-free metric, citing the relevant seminal works. We hope these additions will help readers better understand our approach.
>
> ---
>
> [1] Zhang M, Huang W, Yang B. Interpreting operation selection in differentiable architecture search: A perspective from influence-directed explanations[J]. Advances in Neural Information Processing Systems, 2022, 35: 31902-31914.
>
> [2] J. Turner, E.J. Crowley, M. O’Boyle, A. Storkey, G. Gray, BlockSwap: Fisher-guided block substitution for network compression on a budget, 2019, arXiv preprint arXiv:1906.04113.
>
> [3] N. Lee, T. Ajanthan, P.H. Torr, SNIP: Single-shot network pruning based on connection sensitivity, 2018, arXiv preprint arXiv:1810.02 340.
>
> [4] C. Wang, G. Zhang, R. Grosse, Picking winning tickets before training by preserving gradient flow, 2020, arXiv preprint arXiv:2 002.07376.
>
> [5] H. Tanaka, D. Kunin, D.L. Yamins, S. Ganguli, Pruning neural networks without any data by iteratively conserving synaptic flow 33 (2020) 6377–6389.

---

> > ### Comment · Reviewer_KFhC · 2025-08-06
> > **The response for rebuttal**
> >
> > Thanks for the author's answer, which has basically resolved my question. I will increase my score to 4 points.

---

> > > ### Author Response · Authors · 2025-08-06
> > >
> > > Thank you for your constructive feedback and for raising the score. We are glad our response resolved your concerns and truly appreciate your recognition of our work.

---

### Official Review · Reviewer_G3GX · 2025-06-29

**Clarity:** 3
**Significance:** 2
**Originality:** 3
**Rating:** 4
**Confidence:** 4

**Summary:**

This paper proposes that prior training-free NAS metrics capture only part of a network’s merit and often fail to generalize across different tasks. Simply combining multiple metrics into a single score with fixed weights also can not transfer well. The authors propose PO-NAS, a NAS algorithm that dynamically learns a tailored combination of training-free metrics for each candidate architecture with a weight vector. Then the score is used as a surrogate model to select the optimal architecture. PO-NAS proposes a four step NAS method to combine train-free and training NAS, comprising initialization, a pre-train stage, a Bayesian stage and an evolution stage. Experiments on multiple NAS benchmarks demonstrate that PO-NAS achieves state-of-the-art performance compared with conventional NAS and other hybrid NAS methods.

**Questions:**

- Could you clarify how the per-architecture weight vector $w_A$ is actually optimized in practice? You mention employing Bayesian Optimization for the surrogate – does this mean a Gaussian Process is tuning the weights for each architecture sequentially?
- PO-NAS relies on training architectures for a small number of epochs (10 on CIFAR, 3 on ImageNet) to get the true performance feedback used for weight updates. How well does this short-term performance correlate with the final performance of architectures?
- Ablation on Metrics Used: You integrate six training-free metrics in the surrogate. Did you investigate how important each one is? For example, if you remove a metric or down-weight it, does performance drop noticeably? Conversely, if a new metric is added, is the framework robust to incorporate it?
- Hyperparameter Sensitivity: The paper introduces multiple components to construct the NAS method, and it is necessary to clarify the sensitivity of the final performance to certain key hyperparameters. How sensitive are the results to the choice of $T_e$ and the population size or number of offspring?
- When evaluating the time overhead, will the computational costs of introduced components—such as the architecture encoder and Bayesian surrogate model training—be accounted for?

**Ethical Concerns:**

["NO or VERY MINOR ethics concerns only"]

**Limitations:**

The limitations have been clarified in the weaknesses and questions.

**Paper Formatting Concerns:**

The submission generally follows the NeurIPS 2025 format and guidelines correctly.

**Quality:**

3

**Strengths And Weaknesses:**

**Strengths**:
- The paper is generally well-written and organized. It clearly claims its motivation and core ideas.
- The authors have included a detailed appendix with further experimental details and provided the source codes.
- The experimental evaluation is comprehensive. The authors test PO-NAS on 20 different tasks across three NAS benchmarks, including a variety of domains (image classification, scene recognition, segmentation, etc. in TransNAS-Bench-101).
- The proposed method yields state-of-the-art or near-optimal performance on multiple benchmarks.

**Weaknesses**:
- The proposed approach is quite elaborate, involving multiple components and phases: a graph neural network encoder, a metric predictor, per-architecture weight optimization via a surrogate model, Bayesian optimization guiding architecture selection, and an evolutionary algorithm with custom crossover/mutation. This complexity might make the method hard to understand and tune.
- The complex pipeline proposed in this paper introduces a significant number of additional hyperparameters, but no discussion is provided on how sensitive the final outcome is to these hyperparameters.
- Although the results are strong, in some cases the margins over prior art are not very large. The paper would be stronger if it provided a deeper analysis of current results.

---

> ### Author Rebuttal · Authors · 2025-07-29
>
> Thank you for your meticulous evaluation. We address your concerns as follows:
>
> **Weaknesses**
>
> > The proposed approach is quite elaborate, involving multiple components and phases: a graph neural network encoder, a metric predictor, per-architecture weight optimization via a surrogate model, Bayesian optimization guiding architecture selection, and an evolutionary algorithm with custom crossover/mutation. This complexity might make the method hard to understand and tune.
>
> You have precisely identified the higher complexity of PO-NAS. PO-NAS contains multiple components, but each module is designed to address core challenges in Training-Free NAS:
>
> - Architecture Encoder​​: Introduces architectural topological information to resolve stability issues of training-free metrics across different architectures within the same search space.
> - Metric Predictor​​: Serves as a supervisory signal for the architecture encoder and avoids extensive metric calculations for offspring in evolutionary algorithms.
> - Transformer Surrogate Model​​: Integrates advantages of real training and training-free metrics.
> - ​​Evolutionary Algorithm​​: Combined with preceding components, significantly improves generalizability of existing training-free NAS in large search spaces.
>
> Although PO-NAS has complexity, it is flexibly adjustable and easily implementable. In small search spaces (e.g., NAS-Bench-201), benefiting from the high-speed computation of training-free metrics, PO-NAS can directly enumerate all architectures without evolutionary algorithms, greatly reducing complexity. Nevertheless, complexity remains a non-negligible issue, and we will further optimize this limitation in future work.
>
> > The complex pipeline proposed in this paper introduces a significant number of additional hyperparameters, but no discussion is provided on how sensitive the final outcome is to these hyperparameters.
>
> > Hyperparameter Sensitivity: The paper introduces multiple components to construct the NAS method, and it is necessary to clarify the sensitivity of the final performance to certain key hyperparameters. How sensitive are the results to the choice of $T_{e}$ and the population size or number of offspring?
>
> Thank you for raising the hyperparameter sensitivity concern.
>
> - We must clarify that: While the training of the surrogate model and architectural encoder contains many hyperparameters (e.g., learning rate, batch size), these parameters are constrained by the ​​loss threshold mechanism​​ during training. Moreover, the surrogate model training time constitutes ​​a negligible portion of PO-NAS's total computational cost​​, making its fluctuations in overall time and performance negligible. The hyperparameters are exhaustively documented in Appendix B.2 solely to ensure the reproducibility of PO-NAS; in practice, the vast majority of them have negligible impact on the final performance. For critical parameters (e.g., loss threshold, difference threshold), Appendix D.1 already provides comparative experiments and impact analyses.
>
> -  Additionally, regarding evolutionary algorithm parameters (e.g., $T_{e}$, parent count and offspring size), we fully acknowledge the lack of discussion in the paper and will supplement comparative experiments with results analysis for these hyperparameters in the revised version.
>
> > Although the results are strong, in some cases the margins over prior art are not very large. The paper would be stronger if it provided a deeper analysis of current results.
>
> While PO-NAS exhibits marginal performance improvements in some tasks, this precisely underscores its ​​cross-task robustness​​—across ​​20 differentiated tasks​​ spanning multiple search spaces, PO-NAS consistently ​​maintains stable performance gains which substantially remedies the shortcomings of existing training-free NAS​​. Due to space constraints, we were unable to demonstrate PO-NAS's robustness advantages over more comparable algorithms across diverse task scenarios. In the revised appendix, we will supplement comparative experiments with similar algorithms on TransNAS-bench-101 to better illustrate PO-NAS's core strengths.
>
> **Questions**
>
> > Could you clarify how the per-architecture weight vector $w_\mathcal{A}$  is actually optimized in practice? You mention employing Bayesian Optimization for the surrogate – does this mean a Gaussian Process is tuning the weights for each architecture sequentially?
>
> - As shown in Table 1, we demonstrate the optimization effectiveness of the per-architecture weight vector $w_\mathcal{A}$ by tracking how the global correlation between architectural evaluation scores and true performance on NAS-Bench-201 evolves across search epochs. While we cannot guarantee theoretical optimality for every individual architecture's weights, the significant improvement in overall correlation indicates that our weight allocation strategy performs effectively in most cases, which we hope addresses your concern.
>
> - Regarding the implementation details of Bayesian optimization, PO-NAS employs not Gaussian processes but rather the Transformer-based surrogate model proposed in Section 2.1 to dynamically generate per-architecture weight vectors. Serving dual roles as both the surrogate model and acquisition function in Bayesian optimization, this model learns to adaptively determine optimal weight allocation strategies for distinct architectural features through end-to-end training.
>
> Table 1: PO-NAS on NAS-Bench-201 Kendall rank correlation coefficient versus search epoch reported as mean ± standard deviation over 10 runs.
> | Dataset          | 10 epochs | 20 epochs | 30 epochs | 40 epochs | 50 epochs |
> |---------------|----------|----------|----------|----------|----------|
> | CIFAR-10| 0.276±0.062     | 0.537±0.036     | 0.584±0.022     |  0.611±0.020     |   0.641±0.017     |
> | CIFAR-100 | 0.311±0.051     | 0.617±0.041     | 0.632±0.019     |  0.639±0.019     |   0.647±0.012     |
> | ImageNet16-120 | 0.228±0.082     | 0.494±0.052     | 0.541±0.039     |  0.574±0.035     |   0.612±0.026     |
>
>
> > PO-NAS relies on training architectures for a small number of epochs (10 on CIFAR, 3 on ImageNet) to get the true performance feedback used for weight updates. How well does this short-term performance correlate with the final performance of architectures?
>
> We fully acknowledge the validity of your insightful comments regarding the correlation between short-term training and final performance, which indeed constitutes a key performance bottleneck for PO-NAS, as thoroughly discussed in Appendix C.4.
>
> However, this design stems from our careful consideration of the trade-off between computational efficiency and generalization capability across different search spaces: PO-NAS incorporates both topological structures and training-free metrics rather than relying solely on short-term performance; while in contrast to purely training-free approaches, the true target-space training is crucial for achieving cross-space generalization. The consistently superior performance of PO-NAS across multiple tasks demonstrates that, although the short-term training correlation presents a theoretical limitation, research findings confirm it still maintains a strong correlation with the architecture's final performance [1, 2]. This deliberate trade-off enables PO-NAS to achieve better performance while maintaining both efficiency and generalizability.
>
> > Ablation on Metrics Used: You integrate six training-free metrics in the surrogate. Did you investigate how important each one is? For example, if you remove a metric or down-weight it, does performance drop noticeably? Conversely, if a new metric is added, is the framework robust to incorporate it?
>
> - Regarding metric importance: Our preliminary experiments across multiple datasets revealed significant variability in each metric's impact - the contribution of individual metrics to predictor loss and surrogate model performance fluctuates considerably across different datasets, search spaces, and even among architectures within the same search space. This observation motivated the core design of PO-NAS, as we believe dynamic adaptation of metric importance during search is crucial for achieving robust generalization in training-free NAS.
>
> - Concerning metric quantity: As demonstrated in Appendix D.1, the quantity  of metrics substantially influence PO-NAS's performance. We provide guidelines for adjusting metric quantity based on available computational budget.
>
> - About metric extensibility: To ensure fair comparison with baselines, we deliberately adopted identical metric sets (Appendix B.3) (i.e., grad_norm, snip, grasp, fisher, synflow, and jacob_cov from [3]), isolating the performance improvement to our algorithmic contribution rather than metric advantage. However, preliminary results with enhanced metrics show PO-NAS can achieve further improvements, which we will include in the revised manuscript to demonstrate the framework's full potential.
>
> > When evaluating the time overhead, will the computational costs of introduced components—such as the architecture encoder and Bayesian surrogate model training—be accounted for?
>
> We have thoroughly documented PO-NAS's time evaluation protocol in Appendix B.2, where we account for the computational costs of all components in our time assessment.
>
> It is particularly noteworthy that the overhead associated with both the architecture encoder and Bayesian surrogate model training represents only a negligible fraction when compared to the predominant computational expense of short-term architecture training.
>
> ---
>
> [1] Mellor J, Turner J, Storkey A, et al. Neural architecture search without training[C]//International conference on machine learning. PMLR, 2021: 7588-7598.
>
> [2] Dong X, Yang Y. Nas-bench-201: Extending the scope of reproducible neural architecture search[J]. arXiv preprint arXiv:2001.00326, 2020.
>
> [3] Zero-Cost Proxies for Lightweight NAS, ICLR 2021.

---

> > ### Comment · Reviewer_G3GX · 2025-08-05
> >
> > Thanks to the authors for the detailed response  and the effort put into the rebuttal. This response has largely addressed my concerns. I will maintain my original rating. Thanks again.

---

> > > ### Author Response · Authors · 2025-08-05
> > >
> > > Thank you for reviewing our response. We are pleased it has resolved your concerns and are grateful for your support of our work.

---

### Official Review · Reviewer_8Jw8 · 2025-07-01

**Clarity:** 3
**Significance:** 2
**Originality:** 2
**Rating:** 4
**Confidence:** 5

**Summary:**

The paper notes that individual training-free metrics capture partial architectural features (for e.g. Parameter Count capture just computation, not much about connectivity, etc.). They claim that  combining training-free metrics and adjusting to sensitivities of different architectures to specific metrics is needed. PO-NAS is proposed, combining multiple training-free metrics as scores, and combining them smartly. They further use an EA with predictions from surrogate model to search in large search spaces.

**Questions:**

Aside from evaluations missing relevant work from prior literature, I do not have major concerns, paper is well written, well motivated and does not raise any major red flags.

**Ethical Concerns:**

["NO or VERY MINOR ethics concerns only"]

**Final Justification:**

The authors provided a good rebuttal, with several of my concerns addressed. The only reason I am hesitating to assign a `5: Accept` is because I have reservations about "... moderate-to-high impact on more than one ..." that is attached with the rating, as several components already existed in NAS literature in similar forms, but this is a solid paper.

**Limitations:**

Discussed in weaknesses above, just needs more evaluation.

**Paper Formatting Concerns:**

None, figure captions could be made bigger if possible.

**Quality:**

3

**Strengths And Weaknesses:**

Strengths
- Good idea to combine multiple training-free metrics, train an arch-encoder to predict these metrics and use that for broader NAS.

Weaknesses:
- Combining such metrics has already been studied in MultiPredict, FLAN, TA-GATES. Comparing with such methods is important, as they report results on NB201 as well.
- Comparing encoders such as Arch2Vec and CATE might be insightful, it may especially help at the attention-network where distinguishing architectures is important.
- On macro-spaces, comparing with GENNAPE or relevant discussion is important.

[CATE] Yan, Shen, et al. ‘CATE: Computation-Aware Neural Architecture Encoding with Transformers’. arXiv [Cs.LG], 2021, http://arxiv.org/abs/2102.07108. arXiv.

[Arch2Vec] Yan, Shen, et al. ‘Does Unsupervised Architecture Representation Learning Help Neural Architecture Search?’ NeurIPS, 2020.

[MultiPredict] Akhauri, Yash, and Mohamed S. Abdelfattah. ‘Multi-Predict: Few Shot Predictors For Efficient Neural Architecture Search’. arXiv [Cs.LG], 2023, http://arxiv.org/abs/2306.02459. arXiv.

[TAGATES] Ning, Xuefei, et al. ‘TAGATES: An Encoding Scheme for Neural Network Architectures’. Advances in Neural Information Processing Systems, edited by S. Koyejo et al., vol. 35, Curran Associates, Inc., 2022, pp. 32325–32339, https://proceedings.neurips.cc/paper_files/paper/2022/file/d0ac28b79816b51124fcc804b2496a36-Paper-Conference.pdf.

[FLAN] Akhauri, Yash, and Mohamed S. Abdelfattah. ‘Encodings for Prediction-Based Neural Architecture Search’. arXiv [Cs.LG], 2024, http://arxiv.org/abs/2403.02484. arXiv.

[GENNAPE] Mills, Keith G., et al. ‘GENNAPE: Towards Generalized Neural Architecture Performance Estimators’. Proceedings of the AAAI Conference on Artificial Intelligence, vol. 37, no. 8, Association for the Advancement of Artificial Intelligence (AAAI), June 2023, pp. 9190–9199, https://doi.org/10.1609/aaai.v37i8.26102.

---

> ### Author Rebuttal · Authors · 2025-07-29
>
> We appreciate your close reading and constructive feedback. The points you raised are central to our work, and our answers to each follow in the next sections.
>
> **Weaknesses**
>
> > Combining such metrics has already been studied in MultiPredict, FLAN, TA-GATES. Comparing with such methods is important, as they report results on NB201 as well.
>
> Thank you for the series of papers you recommended. We have carefully read them and confirmed that their experimental settings are highly comparable to PO-NAS and are of great reference value.
>
> To this end, we selected those whose experimental conditions, datasets, and reported metrics most closely match PO-NAS, and we have supplemented direct comparisons on both NAS-Bench-201 (Table 1, Table2) and DARTS  (Table 3). The results demonstrate that PO-NAS effectively couples real training with training-free metrics, achieving superior performance at markedly reduced search costs.
>
> These additional results will help readers gain a more comprehensive understanding of PO-NAS’s true performance and unique advantages, and we will also cite these papers in the revised manuscript for detailed comparative analysis.
>
> Table 1: Comparison of the number of architectures that must be trained for GENNAPE, FLAN$^T_{CAZ}$​, and PO-NAS to attain the target test accuracy on NAS-Bench-201.
> | Algorithm          | GENNAPE[1] | FLAN$^T_{CAZ}$[2] | PO-NAS |
> |---------------|----------|----------|----------|
> | Trained models| 50     | 32     | 20     |
> | Test Accuracy (%) | 93.27     | 93.30     | 94.12     |
>
> Table 2: The mean and standard deviation of the test accuracy(%) of BOHB, arch2vec and PO-NAS under three datasets on NAS-Bench-201.
> | Algorithm          | CIFAR-10 | CIFAR-100 | ImageNet-16-120 | Cost<br>(GPU Seconds) |
> |---------------|----------|----------|----------|----------|
> | BOHB[3]| 93.61±0.52     | 72.37±0.90     | 45.26±0.83     |  12000     |
> | arch2vec-RL[4] | 94.12±0.42     | 73.15±0.78     | 46.16±0.38     |  12000     |
> | arch2vec-BO | 94.18±0.24     | 73.37±0.30     | 46.27±0.37     |  12000     |
> | PO-NAS | 94.12±0.22      | 73.51±0.00     | 46.71±0.12      |  3162     |
>
> Table 3: Comparative results of CATE, arch2vec, and PO-NAS on DARTS Search Space with CIFAR-10
> | Algorithm          | Test Error (%)  | Params (M)  | Search Cost|
> |---------------|----------|----------|----------|
> | CATE-DNGO-LS (small budget)[5]| 2.55 ± 0.08     | 3.5     | 3.3 (GPU days)     |
> | CATE-DNGO-LS (large budget) | 2.46 ± 0.05     | 4.1     | 10.3 (GPU days)    |
> | arch2vec-RL | 2.65±0.05     | 3.3     | 8.3 (GPU days)    |
> | arch2vec-BO | 2.56±0.05     | 3.6     | 9.2 (GPU days)    |
> | PO-NAS | 2.52±0.03      | 3.8     | 3.9 (GPU hours)      |
>
>
> > Comparing encoders such as Arch2Vec and CATE might be insightful, it may especially help at the attention-network where distinguishing architectures is important.
>
> Thank you for this suggestion. We recognize that the original manuscript indeed lacks comparative experiments on encoders, thereby limiting a complete justification of PO-NAS’s effectiveness. The encoder in PO-NAS is supervised by training-free metrics, and validating the reliability of this supervisory signal is essential.
>
> We have carefully examined the encoder papers you recommended; among them, Arch2Vec’s experimental setup on NAS-Bench-201 is most comparable to PO-NAS. To preserve Arch2Vec’s original performance, we retain its official training protocol: supervision by actual architecture performance and obtain its fixed embeddings, and then feed these embeddings into the attention network for architecture discrimination, followed by a direct comparison with PO-NAS. Preliminary results are consolidated in Table 4.
>
> The results show that replacing PO-NAS’s encoder with existing SOTA architecture encoders yields only marginal improvements on certain datasets, while overall performance remains nearly identical. This confirms the excellence of PO-NAS’s encoder and further demonstrates that learning architecture representations via training-free metrics is both feasible and highly efficient, substantially reducing the cost of acquiring topological information.
>
> Table 4: Impact of Different Encoders and Supervisory Signals on PO-NAS Performance (reported by test accuracy (%) on NAS-Bench-201).
>
> | Encoder          | CIFAR-10 | CIFAR-100 | ImageNet-16-120 | Supervisory|
> |---------------|----------|----------|----------|----------|
> | arch2vec | 94.25±0.18     | 73.51±0.00     | 46.49±0.37     |  short-term training (10 epochs)     |
> | PO-NAS | 94.12±0.22      | 73.51±0.00     | 46.71±0.12      |  train-free metrics     |
>
> Once again, we appreciate your valuable comment. We will incorporate the above comparison and cite the relevant work in the revised manuscript, while extending the comparison to additional encoders.
>
> > On macro-spaces, comparing with GENNAPE or relevant discussion is important.
>
> We concur with this point. As noted in our first response, we have already included a comparison with GENNAPE[1], and we will further enrich the revised manuscript with theoretical macro-space analyses between PO-NAS and additional algorithms, accompanied by full citations, to provide readers with a deeper understanding of our contribution.
>
> **Paper Formatting Concerns**
>
> > None, figure captions could be made bigger if possible.
>
> Thank you for your suggestion. We will address this issue in the revised manuscript.
>
> ---
>
> [1] Mills, Keith G., et al. ‘GENNAPE: Towards Generalized Neural Architecture Performance Estimators’. Proceedings of the AAAI Conference on Artificial Intelligence, vol. 37, no. 8, Association for the Advancement of Artificial Intelligence (AAAI), June 2023, pp. 9190–9199.
>
> [2] Akhauri, Yash, and Mohamed S. Abdelfattah. ‘Encodings for Prediction-Based Neural Architecture Search’. arXiv [Cs.LG], 2024, arXiv.
>
> [3] Stefan Falkner, Aaron Klein, and Frank Hutter. BOHB: Robust and efficient hyperparameter optimization at scale. In ICML, 2018.
>
> [4] Yan, Shen, et al. ‘Does Unsupervised Architecture Representation Learning Help Neural Architecture Search?’ NeurIPS, 2020.
>
> [5] Yan, Shen, et al. ‘CATE: Computation-Aware Neural Architecture Encoding with Transformers’. arXiv [Cs.LG], 2021, arXiv.

---

> > ### Comment · Reviewer_8Jw8 · 2025-08-05
> >
> > Thanks to the authors for the clarification, I have raised my score.

---

> > > ### Author Response · Authors · 2025-08-05
> > >
> > > Thank you for reviewing our response and for raising the score. We are glad our replies addressed your concerns and truly appreciate your support of our work.

---

### Official Review · Reviewer_SnuL · 2025-07-03

**Clarity:** 3
**Significance:** 3
**Originality:** 3
**Rating:** 5
**Confidence:** 4

**Summary:**

PO-NAS is a Neural Architecture Search (NAS) algorithm that improves the accuracy and efficiency of training-free NAS methods. Unlike previous approaches that apply a fixed combination of training-free metrics across the whole search space, PO-NAS dynamically optimizes metric combinations for each individual architecture using limited real training data. It integrates multiple metrics, avoids benchmark dependency, and incorporates an evolutionary search strategy guided by a surrogate model. The method effectively balances the speed of training-free NAS with the reliability of training-based evaluations, achieving strong performance across various tasks.

**Questions:**

as my concerns

**Ethical Concerns:**

["NO or VERY MINOR ethics concerns only"]

**Final Justification:**

The authors' responses are reasonable to me, and as a result, I have decided to raise my evaluation to Accept.

**Limitations:**

yes

**Quality:**

3

**Strengths And Weaknesses:**

Strengths:

1. The proposed PO-NAS introduces a novel method that dynamically weights existing zero-cost estimates.
2. Through a well-crafted system design, PO-NAS significantly outperforms the performance of individual zero-cost estimates.
3. The paper is clearly written and easy to follow.

Minor Concerns:
1. The use of both ELU and GELU activation functions appears inconsistent. Is there a specific reason for this combination?
2. I am also curious whether different permutations or combinations of zero-cost estimates have any impact on performance.
3. It would be valuable to discuss the relationships between different zero-cost estimates, for example, whether SynFlow and SNIP might be complementary. Such analysis could further enhance the contribution of the work.

Comment:
Overall, this paper is well-written, with a clear motivation and strong experimental results. Discussing the potential relationships among different zero-cost estimates could further strengthen the contribution. So I recommend borderline acceptance.

---

> ### Author Rebuttal · Authors · 2025-07-29
>
> We sincerely thank you for the time and effort you have invested in reviewing our paper. Your comments and questions are very insightful and highly valuable for us to enhance the paper. Your questions are addressed point-by-point in the following.
>
> **Minor Concerns**
>
> > The use of both ELU and GELU activation functions appears inconsistent. Is there a specific reason for this combination?
>
> The hybrid use of ELU and GELU is a targeted design based on the distinct requirements for processing topological features at different network levels:
>
> - The application of ELU after the GAT layer is motivated by the characteristics of node connectivity relationships in the topological structure. The topological information processed by GAT includes connection objects and operation types between nodes, where certain operation connections may negatively correlate with prediction outcomes and thus require negative-valued features for accurate representation. ELU's smooth negative region property $(a(e^{x} - 1))$ preserves these prediction-significant negative-valued features while preventing the information loss that would occur with ReLU's zero-truncation, which is crucial for accurately modeling the complex relationships between topological structures and training-free metrics.
>
> - The use of GELU after the linear layer serves to enhance the expressive power of high-level topological features. The linearly transformed topological features demand more refined nonlinear processing, where GELU's probabilistic gating mechanism $(x \cdot \Phi(x))$better captures the impact of complex interaction patterns between nodes on prediction metrics, while its smooth properties facilitate hierarchical feature propagation. This combination ensures both complete representation of low-level topological features and improved abstraction of high-level features, representing a carefully considered design solution.
>
> This insightful question has helped us better articulate the rationale behind our activation function choices. We will explicitly address this point in the revised manuscript to improve the interpretability of our design methodology.
>
> > I am also curious whether different permutations or combinations of zero-cost estimates have any impact on performance.
>
> Thank you for raising this insightful question.  In conventional training-free NAS, the permutation or combination of training-free metrics inevitably influences the final ranking quality, because each metric is usually tailored to specific scenarios (e.g., SWAP[2] is only effective for ReLU-based networks). Consequently, a fixed combination can exhibit large performance gaps across different tasks. PO-NAS addresses this limitation through its learnable linear weighting: during the search, the learnable weight vector $w_\mathcal{A}$ dynamically assigns weights to each training-free metric, which is equivalent to implicitly constructing and evaluating a new metric combination at every step without enumerating all possible explicit combinations.
>
> If you were referring to the six training-free metrics adopted in the paper, we must clarify that, to ensure a fair comparison with baselines, we strictly followed the set specified in Appendix B.3 (grad_norm, snip, grasp, fisher, synflow, and jacob_cov from [1]), guaranteeing that any performance gain originates from the PO-NAS algorithm itself.
>
> Nevertheless, incorporating recently proposed training-free metrics (such as SWAP [2] or AZ-NAS [3]) can further improve the average performance of PO-NAS, and we will include these results in the revised manuscript to demonstrate the framework’s extensibility. Finally, as the training budget increases, PO-NAS can explore larger pools of training-free metrics and their combinations, thereby yielding additional performance gains.
>
> > It would be valuable to discuss the relationships between different zero-cost estimates, for example, whether SynFlow and SNIP might be complementary. Such analysis could further enhance the contribution of the work.
>
> We fully acknowledge the importance of analyzing the complementary relationships among training-free metrics.
>
> Different metrics (e.g., NASWOT[4] based on ​​binary activation patterns​​, SNIP[5] relying on ​​gradient distribution​​, and ZenNAS[6] analyzing ​​linear regions​​) exhibit inherent complementarity due to their distinct underlying principles. Combining these metrics enables a comprehensive evaluation of architectural characteristics.
>
> However, these relationships demonstrate significant volatility: cross-task correlations vary drastically, and fluctuations occur across search spaces and even individual architectures within the same space. This non-generalizable volatility constitutes the core motivation for PO-NAS's dynamic metric weighting mechanism​​—which adaptively adjusts weights to meet diverse task and architectural demands while ​​consistently achieving robust performance​​. We will strengthen this argument by adding quantitative experiments in the revised manuscript.
>
> ---
> [1] Zero-Cost Proxies for Lightweight NAS, ICLR 2021.
>
> [2] Peng, Y., A. Song, H. M. Fayek, et al. Swap-nas: Sample-wise activation patterns for ultra-fast nas. In ICLR. 2024.
>
> [3] Lee, J., B. Ham. Az-nas: Assembling zero-cost proxies for network architecture search. In 2024 IEEE/CVF Conference on Computer Vision and Pattern Recognition (CVPR), pages 5893–5903. IEEE Computer Society, 2024.
>
> [4] Mellor J, Turner J, Storkey A, et al. Neural architecture search without training[C]//International conference on machine learning. PMLR, 2021: 7588-7598.
>
> [5] Lee N, Ajanthan T, Torr P H S. Snip: Single-shot network pruning based on connection sensitivity[J]. arXiv preprint arXiv:1810.02340, 2018.
>
> [6] Lin M, Wang P, Sun Z, et al. Zen-nas: A zero-shot nas for high-performance image recognition[C]//Proceedings of the IEEE/CVF international conference on computer vision. 2021: 347-356.

---

> > ### Comment · Reviewer_SnuL · 2025-08-05
> >
> > I sincerely appreciate the authors' detailed reply. Their responses are reasonable to me, and as a result, I have decided to raise my evaluation to Accept. Additionally, I would like to kindly ask the authors to keep their promise. To further refine and elaborate on the relevant sections as indicated. Thx again.

---

> > > ### Author Response · Authors · 2025-08-05
> > >
> > > We sincerely appreciate your time in reviewing our response and updating the score.  We are glad that our response has addressed your concerns and grateful for your  support of our work.

---

### Note · Authors · 2025-08-12

Dear Reviewers and AC,

We sincerely appreciate your valuable feedback, which has significantly enhanced our work.  We have addressed all the concerns and made comprehensive revisions to our paper.

- We provide a more detailed explanation of the design principles behind the architecture encoder and the attention weight network.

- We expand more training-free metrics and further demonstrate these results in the appendix to highlight the scalability of PO-NAS.

- We add and compare more NAS baselines on NAS-Bench-201.

- We conduct new ablation experiments on the architecture encoder and compare it with existing encoders (e.g., arch2vec) to demonstrate
 its reliability.

- We optimize the formatting of titles for all figures and tables in the paper to ensure clarity of charts.

- We conduct more ablation experiments on parameters related to evolutionary algorithms.

- We add detailed descriptions of the precise meanings and theoretical foundations of each training-free metric in the appendix.

- We analyze the optimization results using a sensitivity analysis method similar to DARTS-IM to demonstrate the effectiveness of specific metrics.

We sincerely appreciate the invaluable suggestions from the reviewers:

- The detailed explanation of the design principles of network layers and activation functions (`SnuL`).

- The further argumentation on the complementarity between training-free metrics (`SnuL`, `G3GX`).

- The necessity of discussion and comparison of architecture encoders and macro spaces (`8Jw8`).

- The need to complete more NAS baselines and hyperparameter ablation experiments and discuss the results (`G3GX`, `8Jw8`).

- The addition of theoretical foundations of each training-free metric and comparison analysis of these metrics (`KFhC`).

The reviewers have highly recognized our work in the following aspects:

- The **clear and reasonable** motivation for proposing the method PO-NAS (`SnuL`, `G3GX`).

- The **extensive and rational** experiments conducted to demonstrate the **effectiveness and excellent performance** of PO-NAS (`G3GX`, `KFhC`).

- The **novelty and scalability** of the PO-NAS method (`SnuL`, `8Jw8`, `KFhC`).

- The **clarity** of the paper, which includes detailed experimental details and is easy to follow (`SnuL`, `G3GX`).

---

Once again, we thank all reviewers for your suggestions and recognition. We are pleased to have addressed your concerns and questions and are happy to answer any further questions you may have.

---

### Decision · Program_Chairs · 2025-09-17

**Decision:**

Accept (poster)

**Comment:**

The reviewers all acknowledged the interest of combining multiple metrics in a NAS framework, the effectiveness of the method, and the clarity of the paper. They nevertheless raised some questions regarding design choices for the approach and additional empirical evaluations. The authors' feedback addressed most of the reviewers' concerns, thus resulting in a consensus for acceptance. The AC nonetheless encourages the authors to incorporate their feedback in the camera-ready version of the paper.